



# Benchmarking Ensemble Streamflow Prediction skill in the UK

Shaun Harrigan[1], Christel Prudhomme[1,2,3], Simon Parry[1], Katie Smith[1], and Maliko Tanguy[1]

[1]Centre for Ecology & Hydrology, Wallingford, Oxfordshire, OX10 8BB, UK
[2]Department of Geography, Loughborough University, Loughborough, Leicestershire, LE11 3TU, UK
[3]European Centre for Medium-Range Weather Forecasts (ECMWF), Shinfield Park, Reading, RG2 9AX, UK

*Correspondence to*: Shaun Harrigan (shauhar@ceh.ac.uk)

**Abstract.** Skilful hydrological forecasts at sub-seasonal to seasonal lead times would be extremely beneficial for decision-making in water resources management, hydropower operations, and agriculture, especially during drought conditions. Ensemble Streamflow Prediction (ESP) is a well-established method for generating an ensemble of streamflow forecasts in the absence of skilful future meteorological predictions, instead using Initial Hydrological Conditions (IHCs), such as soil moisture, groundwater, and snow, as the source of skill. We benchmark when and where the ESP method is skilful across a diverse sample of 314 catchments in the UK and explore the relationship between catchment storage and ESP skill. The GR4J hydrological model was forced with historic climate sequences to produce 51-member ensemble of streamflow hindcasts. We evaluated forecast skill seamlessly from lead times of 1-day to 12-months initialised at the first of each month over a 50-year hindcast period from 1965-2015. Results show ESP was skilful against a climatology benchmark forecast in the majority of catchments across all lead times up to a year ahead, but the degree of skill was strongly conditional on lead time, forecast initialisation month, and individual catchment location and storage properties. UK-wide mean ESP skill decays exponentially as a function of lead time with mean squared error skill scores across the year of 0.839, 0.303, and 0.179 for 1-day, 1-month, and 3-month lead times, respectively. However, skill was not uniform across all initialisation months. For lead times up to 1-month, ESP skill was higher than average when initialised in summer and lower in winter, whereas for longer seasonal and annual lead times skill was highest when initialised in autumn and winter months and lowest in April. ESP is most skilful in the south and east of the UK, where slower responding catchments with higher soil moisture and groundwater storage are mainly located; correlation between catchment Baseflow Index (BFI) and ESP skill was very strong ($\rho = 0.896$ at 1-month lead time). This is in contrast to the more highly responsive catchments in the north and west which are generally not skilful at seasonal lead times. Overall, this work provides a scientifically defensible justification for when and where use of such a relatively simple forecasting approach is appropriate in the UK and creates a low cost benchmark against which potential skill improvements from more sophisticated hydro-meteorological ensemble prediction systems can be judged.



# 1 Introduction

Skilful hydrological forecasts at sub-seasonal to seasonal lead times would provide a valuable tool for improved decision making for wide range of sectors such as water resources management (Anghileri et al., 2016), hydropower operations (Hamlet et al., 2002), and agriculture (Letcher et al., 2004), particularly in times of slow onset events such as drought (Simpson et al.,

2016). One of the earliest operational hydrological forecasting methods is Ensemble Streamflow Prediction (ESP). ESP was pioneered in the US at the National Weather Service (NWS) during the 1970s and 1980s as a means of providing ensemble forecasts of streamflow volume for a variety of lead times from 1-day to seasonal and beyond (e.g. Day, 1985; Twedt et al., 1977). Two years of severe drought in California in 1976 and 1977 provided the motivation for such hydrological forecasting developments at the time (Wood et al., 2016b). In the UK, the 2010-2012 drought in England and Wales provided the impetus

for the establishment of the first operational seasonal hydrological forecasting service, the Hydrological Outlook UK (HOUK), that went live in June 2013 (Prudhomme et al., 2017, submitted; forecasts available at: http://www.hydoutuk.net/). ESP is used as one of three hydrological forecasting approaches in HOUK and also feeds into the Environment Agency's monthly 'Water Situation Reports for England', providing forward look ESP forecasts of streamflow for 29 catchments out to a 12-month lead time (https://www.gov.uk/government/collections/water-situation-reports-for-england).

In ESP, historical sequences of climate data (precipitation, potential evapotranspiration, and/or temperature) at the time of forecast are used as forcing to hydrological models, providing a plausible range of representations of the future streamflow states. The source of ESP skill is therefore due to Initial Hydrologic Conditions (IHCs) from antecedent stores of soil moisture, groundwater, snowpack, and channel streamflow itself (Wood et al., 2016a; Wood and Lettenmaier, 2008) which can be detectable up to a year ahead (Staudinger and Seibert, 2014), rather than from skilful atmospheric forecasts. The original

operational concept of the NWS ESP forecasting system was that it was flexible, easy to use, and could be run efficiently using simple conceptual hydrological models (Day, 1985). ESP is still widely used today in operational seasonal hydrological forecasting (e.g. US NWS and HOUK) and as a low cost forecast against which to benchmark potential skill improvements from more sophisticated hydro-meteorological ensemble prediction systems (e.g. Crochemore et al., 2017; Pappenberger et al., 2015; Thober et al., 2015; Wood et al., 2005).

Several studies have established the skill of the ESP method for catchments in particular regions based on carefully constructed hindcast experiments. For example, in the western US, Franz et al. (2003) found ESP forecasts in 14 snow dominated catchments were, on average, skilful (compared to benchmark climatology forecasts) out to at least 7-month lead time, particularly when initialised early in the spring snowmelt season. Wood and Lettenmaier (2008) found that information about IHCs was more important than climate information during the transition between wet and dry seasons in two western

US catchments out to at least a 5-month lead time. For non-snow dominated catchments in the south east of the US, Li et al. (2009) showed that harnessing the long memory of soil moisture and groundwater stores can provide skilful ESP forecasts, as the impact of anomalous dry or wet conditions can take weeks or months to dissipate. Wang et al. (2011) found simple conceptual rainfall-runoff models were able to reliably estimate conditional catchment IHCs in two east Australian catchments,



subsequently producing ESP forecasts of comparable skill to the current operational Bayesian Joint Probability statistical forecast system (BJP, Wang et al., 2009) out to at least 1- and 3-month lead times. More recently, Singh (2016) assessed the potential for long-range ESP forecasting for integrated water management in four catchments (two rainfall dominated and two snowfall dominated) in South Island New Zealand and found ESP to be skilful out to at least a 3-month lead time, with greatest

improvements over climatology forecasts in summer. The skill of ESP has not yet been investigated at the catchment-scale in the UK within a rigorous hindcast experiment and is therefore the focus of this paper.

By definition, a forecast can only be considered *skilful* if it is more accurate against observations than some simpler and/or cheaper reference or *benchmark* forecast (Jolliffe and Stephenson, 2003; Wilks, 2011). Pappenberger et al. (2015) identified three classes of benchmark forecasts commonly used in hydrological forecasting: (i) climatology, used for seasonal forecasting,

(ii) persistence, used for short range forecasting, and (iii) simplified hydrology models, for testing whether more complex models provide useful skill gains. We define the process of *benchmarking* as establishing the skill of a forecasting system (here ESP) against a simpler benchmark forecast across various lead times, forecast initialisation months, and for a large sample of diverse catchments within the study domain. Consequently, the aim of this paper is to establish the skill of the ESP method for forecasting streamflow volumes in the UK at the catchment-scale using (streamflow) climatology as the benchmark

forecast within a rigorous 50-year hindcast study design. Three key research questions emerge:

1. When is ESP skilful, in terms of a wide range of lead times and forecast initialisation months?

2. Where is ESP skilful, in terms of spatial distribution of skilful forecasts both regionally and at the individual catchment-scale across the UK?

3. Why is ESP skilful, in terms of individual catchment soil moisture and groundwater storage capacity?

Section 2 describes the hydroclimatic data used and the selection of catchments, Sect. 3 outlines the methods leading to the generation of ESP hindcasts. Results are presented in Sect. 4 and discussed in Section 5, before key conclusions and avenues for further work are offered in Sect. 6. Details about how to access the ESP hindcast archive used in this study and supplementary data are given in Sect. 7.

**2 Data**

We selected a set of 314 catchments for our ESP evaluation from the UK National River Flow Archive (NRFA; http://nrfa.ceh.ac.uk/) chosen to be representative of the range of UK hydroclimatic conditions and ensuring good spatial coverage (Fig. 1). These catchments include those used for routinely assessing the current and future UK hydrological status (e.g. NHMP, 2017) as well as 132 catchments that are part the new version of the UK Benchmark Network (UKBN2, Harrigan et al., 2017, submitted) that can be considered relatively free from human disturbances such as water abstractions,

urbanisation, and reservoir impacts. Individual details of all 314 catchments are given in supplementary Table S1.

Observed catchment average daily mean streamflow $Q$ ($m^3 s^{-1}$), daily precipitation $P$ ($mm\ d^{-1}$), and daily potential evapotranspiration $ET_p$ ($mm\ d^{-1}$) were extracted for each catchment and are needed for three tasks: i) as input to the



hydrological model calibration (Q, P, and $ET_p$; Sect. 3.1); ii) to generate historic climate sequences (P and $ET_p$, Sect. 3.2) used as forcing to the ESP method; and iii) as forcing to the reference simulation (P and $ET_p$; i.e. proxy observations in Section 3.3).

Q was retrieved from the NRFA over the longest possible period of observed Q across the 314 stations (32 water years; October 1982 – September 2014). P was retrieved from the 1 km gridded CEH-GEAR dataset (Keller et al., 2015; Tanguy et al., 2016) between 1961 and 2015 for the UK. $ET_p$ according to Penman-Monteith for FAO-defined well-watered grass was retrieved from the 1 km gridded CHESS-PE dataset (Robinson et al., 2016, 2017) between 1961 and 2015 for catchments in Great Britain. CHESS-PE does not cover Northern Ireland, so an alternative 5 km $ET_p$ dataset for the UK based on the temperature-based McGuinness-Bordne equation was used instead for these 10 catchments (Tanguy, 2017, in preparation).

Catchment characteristics are summarised in Table 1 for the UK and nine hydroclimate regions as shown in Fig. 1. The distribution of the 314 catchments within the nine regions varies between 10 in Northern Ireland (NI) and 61 in Southern England (SE). Catchment areas range from 4.4 km$^2$ to 9948 km$^2$ with a median area of 181 km$^2$. There is a distinctive hydroclimatic gradient in the UK with wetter more responsive upland catchments in the north and west, and drier lowland catchments in the south and east, some of which drain the principal Chalk, Limestone and Sandstone aquifers. The slow flow contribution from groundwater and other delayed sources, such as lakes, snow, and soil water storage, was characterised using the Base Flow Index (BFI; Gustard et al., 1992) obtained from UK NRFA metadata. BFI ranges between 0 and 1 with values ~0.15-0.35 representative of more responsive rainfall-runoff regimes in the north and west whereas many highly productive Chalk rivers in the south east have a BFI > 0.9. Three regions, Severn-Trent (ST), Anglian (ANG) and SE, have median runoff-ratios (RR) < 0.5 meaning more precipitation is lost to evaporation than runoff in the majority of these catchments. Less than 5 % of catchments have a significant amount of snowfall, defined here following Berghuijs et al. (2014) as catchments with a long-term mean fraction of precipitation falling as snow $\overline{F_s} > 0.15$, and are mainly situated in Eastern Scotland (ES). The range of hydroclimatic characteristics of the catchments used provide a large and diverse set of catchments to benchmark ESP skill.

## 3 Methods

### 3.1 Hydrological modelling

The first of four key methodological steps was to calibrate and evaluate the GR4J (Génie Rural à 4 paramètres Journalier) model (Perrin et al., 2003) used for the generation of streamflow series. It is a daily lumped catchment rainfall-runoff model with a parsimonious structure consisting of four free parameters that require calibration against steamflow observations using daily P and $ET_p$ as input. GR4J has been shown to reliably simulate the hydrology of a diverse set of catchments (Perrin et al., 2003) including temporal transition between wet and dry periods (Broderick et al., 2016), and for the generation of ESP forecasts (e.g. Pagano et al., 2010). The GR4J structure includes a soil moisture accounting reservoir (size controlled with parameter X1 [mm]) with a water exchange function (rate controlled by parameter X2 [mm d$^{-1}$]), and a non-linear routing store to represent base flow (size determined by parameter X3 [mm]), with rainfall-runoff time lags (set in days by parameter X4 [d]) controlled by two unit hydrographs.



GR4J was calibrated using the open source 'airGR' package v1.0.2 in R (Coron et al., 2016, 2017) with the inbuilt calibration optimisation algorithm based on a steepest descent local search procedure and default parameter ranges. The modified Kling-Gupta efficiency ($KGE_{mod}$, Gupta et al., 2009, Kling et al., 2012) applied to root squared transformed flows $KGE_{mod}[sqrt]$ was used as the objective function for automatic fitting, thus placing weight on mid-range flows, rather than high

or low flows. This was decided given ESP forecasts are made across the year during both dry and wet conditions. A split sample test (Klemeš, 1986) was used by dividing the 32 year complete period (CP, water years 1983-2014) of available streamflow observations into two equal 16 year segments for calibration and evaluation: period 1 (P1, water years 1983-1998) and period 2 (P2, water years 1999-2014). Three calibrated GR4J parameter sets were created for each catchment using data from P1, P2, and CP, thus allowing testing of parameter stability between P1 and P2. Model performance against streamflow

observations was evaluated using $KGE_{mod}[sqrt]$ and percent bias (PBIAS) (Gupta et al., 1999) to assess water balance errors.

The UK-wide median ($5^{th}$ and $95^{th}$ percentile) $KGE_{mod}[sqrt]$ is 0.94 (0.83, 0.97) for calibration (CP) and for evaluation 0.92 (0.8, 0.96) and 0.92 (0.78, 0.96) for P1 and P2, respectively (Table 2). Median PBIAS across all catchments over CP is low -0.1 % (-3.7 %, 0.7 %). Overall, GR4J performs well against streamflow observations and parameter sets remain stable across P1 and P2. All streamflow simulations (proxy observations, and benchmark and ESP forecasts) were generated using model

parameter sets calibrated over CP; median and ranges of calibrated parameter values for GR4J X1:X4 across the UK and nine hydroclimate regions are given in Table 2 and for individual catchments in supplementary Table S1 along with respective performance metrics.

## 3.2 Generation of ESP hindcasts from historic climate data

In step 2, Initial Hydrologic Conditions (IHCs) were estimated for each catchment and forecast initialisation date by forcing

the calibrated GR4J model with four years of observed P and $ET_p$ previous to the forecast initialisation date, over the 1961 to 2015 period, thus the first usable forecast date after model spin up is 1 January 1965. Secondly, a 51-member ensemble $m$ of streamflow hindcasts was generated for each forecast initialisation date (first of each month) by forcing GR4J with 51 historic climate sequences (P and $ET_p$ pairs) extracted from 1961 to 2015 out to 12-months lead time at a daily time-step. Each of the 51 generated hindcast time-series were then temporally aggregated to provide a forecast of streamflow volume with seamless

lead times of 1-day to 12-months, resulting in 365 lead times LT per forecast (leap days were removed).

Although it is not possible to create a hindcast experiment under exactly the same conditions experienced in operational mode, effort was made to ensure historic climate sequences did not artificially inflate skill (Robertson et al., 2016) by using leave-3-years-out cross-validation (L3OCV) whereby the 12-month forecast window and the two subsequent years were not used as climate forcings. This was done to account for persistence from known large-scale climate-streamflow teleconnections

such as the North Atlantic Oscillation with influences lasting from several seasons to years (Dunstone et al., 2016). Using the first forecast on 1 January 1965 as an example, 51 sequences of P and $ET_p$ pairs of length 365 days (from 1 January to 31 December) were extracted from observed P and $ET_p$ records between 1961 to 2015, but not for 1965, 1966, or 1967. To keep a 51-member ensemble across all hindcast years, forecasts made in 2013 and 2014 did not have enough data for L3OCV so in



these cases climate sequences from 1961, and 1961 and 1962, respectively were instead removed. The skill of ESP was evaluated over a 50-year hindcast period $N$ between 1965 and 2015 for each of 12 initilisation months $i$ (January to December) and all 365 LT. In total, 600 hindcasts were generated ($N \times i$) with $m = 51$ ensemble members each at LT = 365 lead times across 314 catchments resulting in over $3.5 \times 10^9$ forecast values of streamflow volume in the ESP hindcast archive.

## 3.3 Creation of proxy streamflow observation series

In step 3, a proxy streamflow observation series was produced by forcing the calibrated GR4J model with observed P and $ET_p$ over 1961-2015 with a four year model spin-up. A four year model spin up ensures model states are appropriately stabilised, especially important for slower responding catchments (e.g. in Southern England and Anglian regions). The proxy observation series, the best estimate of streamflow observations given current model and observed meteorological data, is used to evaluate ESP and benchmark forecasts against. It is common to use this approach instead of using direct streamflow observations as it has the advantage of isolating loss of skill to IHCs rather than from model errors and biases (e.g. Alfieri et al., 2014; Pappenberger et al., 2015; Wood et al., 2016a; Yossef et al., 2013).

## 3.4 Evaluation of ESP skill

In step 4, forecast skill is presented as a skill score, which is the improvement over the benchmark forecasts using some measure of accuracy $A$, given generically by Wilks, (2011) in Eq. (1):

$$\text{Skill Score} = \frac{A_{fc} - A_{bench}}{A_{perf} - A_{bench}} \tag{1}$$

where $A_{fc}$ is the accuracy measure of the forecasting system $Q_{fc}$ (here ESP) against observations $Q_{obs^*}$ (here *proxy observations); $A_{bench}$ is the accuracy measure of the benchmark forecast $Q_{bench}$ against $Q_{obs^*}$, and $A_{perf}$ is the value of $A$ in the case of a perfect forecast (typically 1 or 0 depending on metric). For each forecast made over the hindcast period two aspects of ESP forecasts were evaluated: i.) the deterministic skill of the 51-member ESP ensemble mean forecast $Q_{fc\_mean}$ against a deterministic climatology benchmark $Q_{bench\_mean}$ calculated as the 1965-2015 long-term mean proxy streamflow observations for the time of forecast, and ii) the probabilistic skill of the full ESP 51-member ensemble forecast $Q_{fc\_full}$ against a probabilistic climatology benchmark $Q_{bench\_full}$ calculated as the full sample climatological distribution of proxy streamflow observations over 1965-2015 for the time of forecast. Similar to the historic climate forcing sequences in Sect. 3.2, climatology benchmark forecasts (both deterministic and probabilistic) were calculated using L3OCV to account for persistence known to occur for several years in streamflow, particularly during drought (Wilby et al., 2015). It was found in testing that ESP skill was artificially advantaged (disadvantaged) if cross-validation was not carried out in historic climate forcings (benchmark forecasts), in some cases by $\pm 15\%$.

Two common accuracy measures $A$ were chosen for evaluation of ESP. The mean squared error (MSE) (Murphy, 1988), and corresponding skill score (MSESS), was used for evaluating the deterministic $Q_{fc\_mean}$ skill, whereas the continuous rank





probability score (CRPS) (Hersbach, 2000), and corresponding skill score (CRPSS), was used for evaluating the probabilistic skill of $Q_{fc\_full}$. The Ferro et al. (2008) ensemble size correction for CRPS was applied to account for differences between $Q_{fc\_full}$ (period 1961-2015 → L3OCV → $n = 51$) and $Q_{bench\_full}$ (period 1965-2015 → L3OCV → $n = 47$), as done in evaluation of hydrological ensemble forecasting elsewhere (e.g. Crochemore et al., 2017). Calculation of skill scores were

undertaken using the open source 'easyVerification' package v0.4.2 in R (MeteoSwiss, 2017). For both MSESS and CRPSS, a skill score of 1 indicates a perfect forecast, a Skill Score > 0 shows the ESP forecast is more skilful than the benchmark, a Skill Score = 0 shows ESP is only as accurate as the benchmark, and a Skill Score < 0 warns that ESP is inferior to the benchmark forecast. The MSESS and CRPSS were applied to each of the 314 catchments for each of the 12 initialisation months and 365 lead times for each year over the 50-year hindcast period.

**4 Results**

Results are presented in the following order: First, ESP skill is shown for all 365 lead times (LT), then by forecast initialisation month for a sample of eight representative LTs commonly used in operational hydrological forecasting (i.e. short (1- and 3-days), extended (1- and 2-weeks), monthly (1-month), seasonal (3- and 6-months), and annual (12-months)). Second, the spatial distribution of ESP skill is shown, both averaged across the UK and each of the nine hydroclimate regions, then for

individual catchments to explore sub-region heterogeneity. Third, the relationship between catchment soil moisture and groundwater storage and ESP skill is assessed.

Reducing accuracy of a forecast to a single skill metric value is abstract and difficult to interpret. Throughout the results and discussion sections skill score values are assigned qualitative descriptions according to degree of skill: Very High [0.8, 1]; High [0.6, 0.8); Moderate [0.4, 0.6); Low [0.2, 0.4); Very Low (0, 0.2); No Skill (including negative skill) [-∞, 0]. Five

example 1965-2015 hindcast time-series with skills ranging from very high to negative skill are visualised in Fig. 2 and will act as a graphical reference in the remainder of the paper to aid interpretation of skill.

**4.1. Timing of ESP skill**

**4.1.1 Lead time**

UK-wide mean ESP skill across all catchments and initialisation months decays exponentially as a function of lead time

for both the MSESS and CRPSS metrics (Fig. 3). Mean MSESS (CRPSS) from short (1-day) to extended (2-week) lead times ranges from 0.839 (0.751) to 0.405 (0.296), and across monthly, seasonal and annual lead times from 0.303 (0.205), 0.173 (0.108), to 0.069 (0.038), respectively. There is large spread around mean skill scores for any lead time, depicted by the semi-transparent 5th and 95th percentile bands across the 314 catchments in Fig. 3. For example, at a 2-week lead time the MSESS (CRPSS) values are bound between 0.175 (0.113) and 0.875 (0.708). Skill scores for the deterministic ESP ensemble mean

(measured by MSESS) are systematically higher than those for probabilistic skill (measured by CRPSS) for all lead times (by





on average 0.055 skill score points, up to a maximum of 0.223) but the range of CRPSS values is narrower compared to MSESS. Lower CRPSS skill scores compared to MSESS are expected as the CRPSS metric is penalised for lower forecast sharpness (i.e. higher ensemble spread; Wilks, 2011), which will be particularly prominent for forecasts with low IHCs influence (compare ensemble spread in Fig. 2a with Fig. 2d, as discussed further in Sect. 5).

### 4.1.2 Initialisation month

ESP skill varies depending on initialisation month, and the time-of-year with highest and lowest skill is conditional on the lead time. Figure 4 shows skill scores for initialisation months January to December for short and extended lead times (LTs) as summarised by boxplots across all catchments. Skill scores for these four sample LTs (1-day, 3-day, 1-week, and 2-week) are higher in summer months (June, July, August) and highest in July, whereas lower skill scores are seen for winter (December, January, February) and autumn (September, October, November), with lowest skill scores in either January or October. For the four sample monthly to annual LTs (Fig. 5), skill scores are also highest for the 1-month forecasts when initialised in July, however patterns are reversed for 3-month, 6-month, and 12-month LTs, where the highest forecast skills are for wetter months October, January, and November, respectively. All four monthly, seasonal, and annual LTs have lowest skill scores when initialised in April, which in the UK is a transition month between wetter winter months and drier spring/summer months. Although CRPSS values are systematically lower than MSESS, the pattern of most/least skilful initialisation months remains consistent.

The decay in skill with LT as shown in Fig. 3 also occurs across all initialisation months (Figs. 4 and 5). Whilst mean ESP skill tends towards zero for longer LTs, there are many catchments with much higher skill scores than average. For example, at a 1-month LT initialised in July the average UK-wide ESP skill is moderate (MSESS = 0.303), but 42 catchments have high skill (MSESS > 0.6) and 34 have very low skill (MSESS < 0.2); the 12-month LT January ESP forecast in the Lambourn at Shaw (Southern England SE) is more skilful (MSESS = 0.721) than 1-week January forecasts in over 92 % of catchments.

## 4.2 Spatial distribution of ESP skill

### 4.2.1 UK hydroclimate regions

Figure 6 shows a heatmap of mean ESP skill across initialisation months for the UK, and for nine hydroclimate regions using the MSESS metric. The same patterns are found for CRPSS (not shown). ESP skill has a prominent spatial pattern across the UK consistent over shorter and longer LTs. Least skilful UK regions are Western Scotland (WS), North-west England & North Wales (NWENW), and Northern Ireland (NI), whereas Severn-Trent (ST), Anglian (ANG), and Southern England (SE) are most skilful. Using a 1-week LT as an example, MSESS for SE (0.685) is almost 50 % higher than for WS (MSESS = 0.358). All regions are, on average, skilful out to 1-month LT, but by 3-months skill is very low in WS, NI, NWENW and South-west England & South Wales (SWESW); at LTs up to 6- and 12-monhts ST, ANG, and SE are the only regions to remain skilful, as a whole.



### 4.2.2 Catchment-scale

There is considerable sub-region heterogeneity when skill scores for individual forecasts at the catchment-scale are examined. MSESS values are mapped in Fig. 7 for all 314 catchment locations for a sample of four LTs (ranging from extended to annual) and three initialisation months (January, April, and July). Although WS is considered a low skill region overall at a

1-week LT in Fig. 6 (i.e. MSESS = 0.358), moderate to very high skill forecasts can be made for some catchments at different times of the year. For example, a July 1-week LT forecast (Fig. 7c) is WS can have moderate skill (MSESS > 0.4) for over 60 % of the 34 catchments or even high skill (MSESS > 0.6) for 20% of the catchments; with a maximum MSESS of 0.834 (Inver at Little Assynt). In all regions, almost all individual catchments are more skilful than the reference climatological forecast (i.e. MSESS > 0) for up to extended LTs (i.e. Fig. 7a-c).

Sub-region heterogeneity is much more apparent for monthly, seasonal, and annual LTs (Fig. 7d-l). As in Fig. 6, skill decays at different rates depending on region and lead time, but also initialisation month. However, the finer spatial information in Fig. 7 shows that skill decays towards zero at vastly different rates for individual catchments even within the same region. For example, despite low average skill of January 12-month LT forecasts in Southern England E (MESS = 0.242), 25 % of catchments have modest to high skill. In April, when monthly and longer LTs forecasts are the least skilful UK-wide (i.e. Fig.

5), 36 and 29 % of catchments in SE and ANG regions have moderate to highly skilful forecasts at monthly and seasonal LTs, respectively (Fig. 7e and h). Sub-region heterogeneity is perhaps the most prominent for the Thames basin in SE, with a low April 3-month LT forecast for the Thames at Kingston (MSESS = 0.369, size = 9948 $km^2$), but contrasting skills for two of its sub-catchments with very high skill for the Lambourn at Shaw (MSESS = 0.859, size = 234 $km^2$) and effectively no skill for the Mole at Kinnersley Manor (MSESS = 0.014, size = 142 $km^2$).

**4.3. Relationship between catchment storage and ESP skill**

The relationship between the two calibrated GR4J catchment storage parameters, X1 (soil store capacity [mm]) and X3 (groundwater store capacity [mm]), BFI, and ESP skill (MSESS) for $n$ = 314 individual catchments is shown in the scatterplot matrix in Fig. 8 using the non-parametric Spearman's rank correlation coefficient $\rho$. Both X1 and X3 are strongly positively (non-linearly) correlated with BFI ($\rho$ = 0.757 and $\rho$ = 0.738, respectively); catchments with highest BFIs tend to have much

higher than average soil moisture and groundwater storage capacity. The BFI is also very strongly positively correlated with ESP skill ($\rho$ = 0.896). The 1-month LT forecast skill (MSESS) averaged across initialisation months is used to demonstrate this, but similar results are found over the range of lead times and individual initialisation months (not shown). Forecasts in the most responsive catchments (BFI ≤ 0.35, 20 % of catchments) have on average very low skill (MSESS = 0.128) whereas in the slowest responding catchments (BFI ≥ 0.9, 5 % of catchments) have very high skill (MSESS = 0.941).





## 5 Discussion

Overall, the ESP method is found to be skilful when benchmarked against climatology in the UK, but the degree of skill is dependent on lead time, initialisation month, and individual catchment location and storage properties.

### 5.1 When is ESP skilful?

UK-wide ESP skill for short lead times (out to 3-days) is on average highly skilful (MSESS > 0.6) and for extended lead times (out to 2-weeks) moderately skilful (MSESS > 0.4). Mean ESP skill decays exponentially with increasing lead time so skill is on average much lower for monthly, seasonal, and annual lead times, as expected. However, the magnitude of skill is not uniform across the 12 forecast initialisation months. ESP skill for short, extended, and monthly lead times is higher than average when initialised in summer months and lower than average for winter and autumn months. Svensson (2016) also found

highest statistical persistence forecast skill across the UK when initialised in summer (highest in July at a 1-month lead time). This is consistent with Li et al. (2009) and Shukla and Lettenmaier (2011) who found soil moisture Initial Hydrologic Conditions (IHCs) contributed to greater skill for forecasts initialised in the warmer summer season than the cold winter season in the south east of the US due to drier initial moisture states in summertime, up to a 1-month lead time. Similarly, Staudinger and Seibert (2014) found drier initial soil moisture was connected to longer persistence in all seasons but winter in Switzerland.

Soil Moisture Deficits (SMDs) are also highest in summer in the UK, peaking in July, and lowest in winter (based on UK Met Office MORECS dataset (Hough and Jones, 1997) over 1961-2015). This could help explain why up to 1-month LT hydrological forecasts initialised in summer months using IHCs alone (e.g. ESP) are more skilful than if initialised in winter in the UK. Higher summer ESP forecast skill could be capitalised upon operationally given seasonal climate predictability over Northern Europe is notoriously challenging for summer rainfall (e.g. Weisheimer and Palmer, 2014).

In contrast, ESP skill at seasonal and annual lead times is higher than average when initialised in winter and autumn months, and lowest in April. However, these higher skills occur in catchments with higher BFIs, suggesting that perhaps groundwater from large slowly responding aquifers is the source of ESP skill at these longer lead times. This is supported by Wood and Lettenmaier (2008) who found that baseflow dominates hydrological persistence in winter in the Rio Grande River in the US. Staudinger and Seibert (2014) also found for simulations initialised in winter, wetter initial conditions lead to longer

persistence, although they note it was difficult to separate the relative influences from snow and aquifer memory. Lower longer-range forecast skill for spring initialisation in the UK was also found by Svensson (2016) for a 3-month LT based on statistical streamflow persistence forecasts. However, there are few seasonal hydrological hindcast studies for the UK that have also assessed skill at longer than 3-month lead times to compare results. Spring in the UK is characterised as a transition season between the wetter autumn and winter and drier summer, in which groundwater recharge no longer occurs and baseflow begins

its recession. Factors that might contribute to lower skilled forecasts initialised in spring include potentially higher uncertainty in IHC storage states and larger variability in rainfall across the forecast window (i.e. from late spring to early autumn). Further work should endeavour to attribute the lack of skill during transition seasons but this is outside the scope of this paper.





## 5.2 Where is ESP skilful?

The skill of ESP is also not uniformly distributed in space. Least skilful hydroclimate regions within the UK are situated in the north and west (WS, NWENW, and NI) whereas the most skilful are situated in the south and east (ST, ANG, and SE) across all lead times studied. This prominent spatial pattern was also noted, among others, by Svensson et al. (2015) and

Svensson (2016) using statistical persistence forecasting and Bell et al. (2017, submitted) using a gridded national hydrological model. These space-time patterns are also apparent in skill maps of individual catchments (i.e. Fig. 7), although there is marked sub-region heterogeneity, as demonstrated using the Thames basin. The slow responding Lambourn at Shaw sub-basin (BFI = 0.97) is very highly skilful compared to the fast responding Mole at Kinnersley Manor catchment (BFI = 0.39).

## 5.3 Why is ESP skilful?

The most skilful ESP regions of the UK are also those that are underlain by the UK's principal aquifers (Fig. 1). Catchments with larger calibrated soil moisture and groundwater storage capacity parameters in GR4J, which are also situated in ST, ANG, and SE (Table 2), tend to have a higher baseflow index (BFI). The BFI is therefore interpreted broadly as an integrated index of catchment soil moisture and groundwater storage capacity here and is inferred to be responsible for modulating ESP skill - catchments with higher storage are more skilful with skill decaying at a much slower rate with increasing lead time, compared

to catchments with low storage capacity. For example, the ensemble mean forecast for the Lambourn remains on average moderately skilful (i.e. MSESS > 0.4) until a lead time of 309 days, while the Mole drops below the moderately skilful threshold at a lead time of just 6 days.

These findings are consistent with current physical understanding of sources of ESP skill in non-snow dominated catchments in the literature. Water storage within the soil introduces a memory effect whereby anomalous dry or wet conditions

can take weeks or months to be 'forgotten' (Ghannam et al., 2016; Li et al., 2009), and the slow transformation of precipitation to streamflow in catchments with productive aquifers in the south east of the UK leads to temporal streamflow dependence for up to a season ahead, and longer (Chiverton et al., 2014). Although it is encouraging that GR4J storage parameter values (X1 and X3) appear to show some physical realism, a note of caution is needed as GR4J is not a physics-based hydrological model, nor is it guaranteed that these results are directly transferable to any lumped catchment hydrological model. It has also been

noted that the BFI is influenced by many other factors such as lake and snow storage (Parry et al., 2016), therefore a more detailed examination of the physical hydrogeological controls on catchment BFI, such as in Bloomfield et al. (2009) for the Thames, is needed at a national scale.

The ESP method was originally developed and tested in the snow dominated catchments of the western US with particular strength in forecasting spring snow melt driven streamflow volumes (e.g. Franz et al., 2003; Wood and Lettenmaier, 2008).

Because the source of ESP is from IHCs, and because individual catchments will have different relative contributions of IHC sources (e.g. snow, soil moisture, and groundwater), ESP skill must be assessed using a large-sample of diverse catchment types and sizes for each region it is being applied in (e.g. Yossef et al., 2013). The present study adds to the broader international



literature in benchmarking ESP skill in non-snow dominated catchments. In particular, results show that IHCs in catchments with large soil moisture and groundwater storage provide skill up to at least a year ahead. It must however be acknowledged that the UK is not completely snow-free. Just under 5 % of catchments studied have a significant snow contribution (i.e. $\bar{F}_s >$ 0.15) located mainly in upland areas of Eastern Scotland (ES) (see Fig. 1). In the present experimental set-up, snow

accumulation and melt processes were not represented within the GR4J model. This would explain why ES has the lowest GR4J model performance for the reference simulation of all regions (Table 2). In addition, the worst performing forecast in the entire ESP hindcast archive is the 3-month LT April forecast for the Dee at Park with a negative MSESS = -0.273 (see Fig 2e). In this instance both the ESP forecast and the proxy streamflow observations (or perfect model) in which the forecast is evaluated against were not a good enough representation of reality.

ESP in its traditional form as used here provides the *lower limit* of streamflow forecasting skill in the absence of skilful atmospheric forecasts (Pagano et al., 2010) or improved hydrological process representation (e.g. snow). As such, ESP assumes near total uncertainty about future rainfall; when there is limited to no influences of IHCs on streamflow generation (e.g. highly responsive catchments), the ESP ensemble mean and spread defaults to climatology (see Fig. 2c). Given the known influence of the NAO on rainfall and therefore streamflow in the UK, particularly in the north and west for winter (e.g. Svensson et al.,

2015), there is potential for an NAO-conditioned ESP method to be developed. This would involve sub-sampling historic climate sequences used to force ESP based on year's most similar to NAO conditions at the time of forecast. Beckers et al. (2016) developed an ENSO-conditioned ESP method for three test sites in the US Pacific Northwest and found skill improvements in the order of 5 to 10 %, and showed the need to include a weather resampling technique to account for the unavoidable reduction in ensemble size. Overall low ESP forecast performance and sharpness in highly responsive catchments

in the north and west would be expected to improve (e.g. Fig. 2c and d) with incorporation of information that reduces rainfall forcing uncertainty at all lead times but particularly seasonal, whether from ensemble sub-sampling or inclusion of skilful atmospheric forecasts..

## 6 Conclusions

Ensemble Streamflow Prediction (ESP) has a rich history internationally as a low cost and efficient ensemble hydrological

forecasting system used operationally across a range of lead times. The ESP method using simple lumped conceptual hydrological models is currently one of three methods used within the operational seasonal hydrological forecasting service UK Hydrological Outlook (HOUK) and also feeds into the Environment Agency's monthly 'Water Situation Report for England'. However, the skill of ESP at the catchment-scale under a rigorous hindcast experiment for a large-sample of diverse catchments across the UK had not previously been investigated.

We conclude that ESP is skilful against a climatology benchmark forecast in the majority of catchments across all lead times up to a year ahead, but the degree of skill is strongly conditional on lead time, forecast initialisation month, and individual catchment location and storage properties. In summary:





- ESP skill decayed exponentially with increasing lead time but catchments with larger soil moisture and groundwater storage capacity decayed at a much slower rate, resulting in the possibility of low to moderate skill forecasts based on Initial Hydrological Conditions (IHCs) alone even at a 12-month lead time for some catchments.

- For short (1- to 3-days), extended (1- to 2-weeks), and monthly forecasts, skill was highest when initialised in summer months, and lowest in winter and autumn months.

- For seasonal (3- to 6-months) and annual forecasts, skill was highest when initialised in winter and autumn months, but only for catchments with high soil moisture and groundwater storage (i.e. high Baseflow Index). Longer range forecast skill was lowest when initialised in spring, particularly April, which is likely due to the complex interplay of hydrological and climatological processes involved during the transition from wetter autumn and winter conditions to drier spring and summer conditions and needs to be explored further.

- ESP is most skilful in the south and east of the UK, where slower responding catchments with higher soil moisture and groundwater storage are mainly located. This is in contrast to the more highly responsive catchments in the north and west which are generally not skilful at seasonal lead times. However, substantial sub-region heterogeneity was observed and skilful ESP forecasts are still possible at the individual catchment-scale despite when the region as a whole has low skill.

We show that simple lumped conceptual rainfall-runoff models (here using GR4J) are able to be used to produce skilful ESP forecasts at short to annual lead times in the UK. This hindcast experiment provides a scientifically defensible justification for when (lead time and initialisation month) and where (region and catchment types) use of such a relatively simple forecasting approach is appropriate. Currently, ESP is only used operationally in the UK at seasonal and annual lead times in England and Wales. This skill evaluation has shown that much higher skills are possible for short (1- to 3-days) and extended (1- to 2-weeks) lead times in all regions across the UK and opens the potential for applying ESP as a low cost and efficient catchment-scale ensemble hydrological forecasting system in a wider context.

Finally, most ensemble hydrological forecasting systems are benchmarked against an arguably too simplistic climatology benchmark forecast which is not particularly challenging to beat. Pappenberger et al. (2015) calls this 'naïve skill' and argues that a forecasting system can only be classified as having 'real skill' when it performs better than a 'tough to beat' lower cost benchmark forecast system. The ESP hindcast archive derived and presented here in itself provides such a 'tough to beat' simplified hydrology model benchmark in which the potential value of improvements from more sophisticated forms of ESP (e.g. incorporation of snow processes, sub-sampling historic climate) or more complex and expensive hydro-meteorological ensemble forecasting systems can be judged. When and where ESP cannot provide skilful streamflow forecasts provides an opportunity to benchmark the degree to which recent improvements in seasonal prediction of UK regional rainfall (e.g. Baker et al., 2017, submitted) leads to improvements over using IHCs alone (i.e. our ESP method), and is the focus of future work.




## 7 Data availability

The ESP hindcast archive (~60 GB) and the 'UK Hydroclimate Regions' shapefile are available from the lead author or by email to the NRFA (nrfa@ceh.ac.uk). Supplementary Table S1 includes metadata for all 314 catchments as well as data used to generate Table 1 and 2 and Fig. 8 for others to explore.

## Acknowledgements

This work was funded by the UK NERC National Capability National Hydrological Monitoring Programme (NHMP) and NERC-funded Improving Predictions of Drought for User Decision-Making (IMPETUS) project (NE/L010267/1). Statistical analyses and graphics were implemented in the open-source R programming language. Streamflow data and metadata are from the NRFA and MORECS dataset from the UK Met Office. We thank Cecilia Svensson for fruitful discussions about this work.

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



**Table 1.** Summary statistics of eight catchment characteristics for the UK and nine hydroclimate regions shown in Fig. 1. The median across n catchments within each region is given with the 5th and 95th percentile ranges in brackets. Area, Median elevation, and Base Flow Index (BFI) were retrieved from the UK NRFA. Mean annual Q, P, and $ET_P$ were calculated over water years 1983 to 2014 using data in Sect. 2. RR is the runoff ratio and $\overline{F_s}$* is the long-term (water years 1983-2014) mean fraction of precipitation that has fallen as snow.

| Region | n | Area (km²) | Median elevation (m a. s. l.) | BFI (-) | Mean annual Q (mm yr⁻¹) | Mean annual P (mm yr⁻¹) | Mean annual $ET_p$ (mm yr⁻¹) | RR $\overline{Q}/\overline{P}$ (-) | $\overline{F_s}$ (-) |
|---|---|---|---|---|---|---|---|---|---|
| UK | 314 | 181 (27; 1844) | 180 (60; 438) | 0.5 (0.27; 0.89) | 595 (162; 1839) | 1031 (648; 2202) | 504 (400; 542) | 0.59 (0.24; 0.87) | 0.03 (0.01; 0.14) |
| WS | 34 | 240 (63; 1751) | 271 (144; 471) | 0.32 (0.2; 0.61) | 1118 (701; 2851) | 1466 (1096; 3147) | 430 (391; 477) | 0.74 (0.62; 0.9) | 0.06 (0.03; 0.13) |
| ES | 44 | 281 (72; 2709) | 303 (100; 596) | 0.52 (0.34; 0.67) | 674 (340; 1494) | 1036 (784; 1969) | 430 (387; 481) | 0.62 (0.44; 0.84) | 0.09 (0.06; 0.21) |
| NEE | 30 | 344 (11; 1910) | 264 (88; 449) | 0.43 (0.26; 0.82) | 559 (344; 1054) | 1037 (757; 1462) | 486 (455; 516) | 0.57 (0.44; 0.83) | 0.07 (0.04; 0.09) |
| ST | 25 | 198 (48; 6345) | 145 (87; 312) | 0.56 (0.41; 0.79) | 392 (209; 844) | 858 (670; 1311) | 511 (493; 528) | 0.46 (0.31; 0.68) | 0.03 (0.02; 0.05) |
| ANG | 33 | 99 (23; 1540) | 82 (33; 132) | 0.56 (0.25; 0.88) | 183 (128; 254) | 655 (600; 716) | 535 (528; 551) | 0.27 (0.21; 0.36) | 0.03 (0.03; 0.04) |
| SE | 61 | 119 (19; 1073) | 105 (43; 187) | 0.64 (0.23; 0.96) | 361 (146; 584) | 861 (654; 1051) | 529 (520; 540) | 0.42 (0.2; 0.64) | 0.02 (0.01; 0.03) |
| SWESW | 45 | 179 (28; 916) | 207 (77; 379) | 0.5 (0.37; 0.67) | 1004 (518; 1551) | 1400 (1011; 1973) | 519 (495; 537) | 0.69 (0.52; 0.83) | 0.01 (0; 0.03) |
| NWENW | 32 | 112 (30; 1094) | 210 (108; 360) | 0.35 (0.27; 0.58) | 1154 (390; 2102) | 1529 (884; 2429) | 478 (457; 514) | 0.75 (0.44; 0.91) | 0.04 (0.02; 0.05) |
| NI | 10 | 230 (68; 1235) | 140 (90; 172) | 0.38 (0.33; 0.5) | 688 (533; 1206) | 1111 (917; 1565) | 475 (466; 488) | 0.63 (0.57; 0.77) | 0.01 (0; 0.02) |

5  * $\overline{F_s}$ calculated using the CemaNeige snow-accounting module (Valéry et al., 2014) within the airGR applied to the GR4J model





**Table 2.** Summary statistics of GR4J calibrated parameters and performance metrics for the UK and nine hydroclimate regions shown in Fig. 1. The median across n catchments within each region is given with the 5th and 95th percentile ranges in brackets.

| Region | n | GR4J X1 (mm) | GR4J X2 (mm d$^{-1}$) | GR4J X3 (mm) | GR4J X4 (d) | Cal (CP) KGE$_{mod}$[sqrt] (-) | Cal (CP) PBIAS (%) | Eval (P1) KGE$_{mod}$[sqrt] (-) | Eval (P2) KGE$_{mod}$[sqrt] (-) |
|---|---|---|---|---|---|---|---|---|---|
| UK | 314 | 250 (78; 955) | -0.1 (-4.2; 0.8) | 40 (12; 380) | 1.3 (1; 2.6) | 0.94 (0.83; 0.97) | -0.1 (-3.7; 0.7) | 0.92 (0.8; 0.96) | 0.92 (0.78; 0.96) |
| WS | 34 | 128 (44; 364) | 0 (-0.5; 0.7) | 26 (14; 135) | 1.2 (1.1; 2.1) | 0.93 (0.83; 0.96) | 0.2 (-2.2; 1.2) | 0.92 (0.82; 0.95) | 0.91 (0.8; 0.95) |
| ES | 44 | 296 (113; 523) | 0 (-0.7; 0.8) | 43 (18; 104) | 1.2 (1.1; 1.8) | 0.9 (0.74; 0.94) | -0.5 (-2.2; 0.4) | 0.88 (0.74; 0.94) | 0.88 (0.71; 0.94) |
| NEE | 30 | 277 (79; 499) | 0 (-1.1; 0.7) | 24 (12; 109) | 1.3 (1.1; 2.3) | 0.92 (0.87; 0.95) | -0.2 (-7.1; 0.4) | 0.91 (0.83; 0.94) | 0.9 (0.78; 0.93) |
| ST | 25 | 345 (142; 1169) | -0.5 (-1; 0.5) | 44 (18; 153) | 1.4 (1.1; 2.7) | 0.96 (0.88; 0.97) | 0.2 (-1.6; 0.7) | 0.93 (0.83; 0.96) | 0.92 (0.8; 0.96) |
| ANG | 33 | 286 (128; 773) | -0.8 (-4.5; -0.1) | 28 (5; 371) | 1.5 (1.2; 2.7) | 0.92 (0.86; 0.95) | -0.2 (-8.7; 1.4) | 0.88 (0.82; 0.94) | 0.88 (0.81; 0.94) |
| SE | 61 | 400 (163; 1868) | -0.7 (-17.1; 0.8) | 83 (6; 681) | 1.4 (1; 9.4) | 0.95 (0.88; 0.97) | -0.1 (-4.6; 0.4) | 0.93 (0.82; 0.96) | 0.92 (0.8; 0.96) |
| SWESW | 45 | 205 (82; 463) | 0.1 (-1; 1) | 80 (29; 171) | 1.2 (0.9; 1.7) | 0.97 (0.94; 0.97) | -0.3 (-1.1; 0.3) | 0.94 (0.86; 0.97) | 0.94 (0.85; 0.97) |
| NWENW | 32 | 141 (60; 480) | 0.2 (-0.6; 0.8) | 36 (19; 134) | 1.2 (1.1; 1.8) | 0.95 (0.93; 0.97) | 0 (-0.5; 0.4) | 0.95 (0.88; 0.96) | 0.94 (0.87; 0.96) |
| NI | 10 | 146 (70; 244) | 0.2 (-0.1; 0.3) | 23 (16; 37) | 1.4 (1.1; 1.9) | 0.93 (0.91; 0.96) | -0.1 (-1; 0.9) | 0.93 (0.86; 0.95) | 0.93 (0.86; 0.95) |



**Figure 1:** Location of 314 gauging stations (red dots) and catchment boundaries (black lines) with upland areas (shaded in grey) and principal aquifers (shaded in pale yellow). UK Hydroclimate Regions, derived from grouping smaller UK hydrometric areas, are shown inset.







**Figure 2:** Five example 1965-2015 hindcast time-series in which skill scores (MSESS and CRPSS) range from very high (a) to negative skill (e). The red line is the 51-member ESP ensemble mean, black line the proxy observed streamflow (also known as a perfect forecast), semi-transparent blue dots show the ensemble spread for each forecast year, and the dashed horizontal black line mean proxy observed streamflow (analogous to a deterministic climatology benchmark forecast, although not cross-validated here as was done in calculation of skill scores (i.e. simply the same value repeated each year)).





**Figure 3:** UK-wide mean ESP skill scores across all 314 catchments and 12 forecast initialisation months for all 365 lead times (LTs) for both MSESS (blue line) and CRPSS (red line) skill metrics. The range of skill scores across catchments at each LT is shown by semi-transparent 5th and 95th percentile bands. Vertical lines represent eight commonly used operational forecasting LTs from short (days) to annual (12-months).







**Figure 4:** UK-wide ESP skill scores across 314 catchments for each of the 12 forecast initialisation months for four short and extended lead times. Blue (red) boxplots summarise MSESS (CRPSS) values with the black line representing the median, and boxes the interquartile range (IQR); whiskers extend to the most extreme data point, which is no more than 1.5 times the IQR from the box, and grey circles are outliers beyond this range.



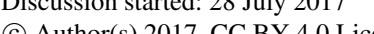

**Figure 5:** As in Fig. 4 but for four monthly, seasonal, and annual lead times.



**Figure 6:** Heatmap of mean ESP skill across all 12 forecast initialisation months for the UK and for each of the nine hydroclimate regions ordered from least to most skilful (horizontal axis) at eight sample lead times (vertical axis). Skill is given by the MSESS with darker (lighter) shades showing higher (lower) skill; individual mean skill values are shown within each cell.





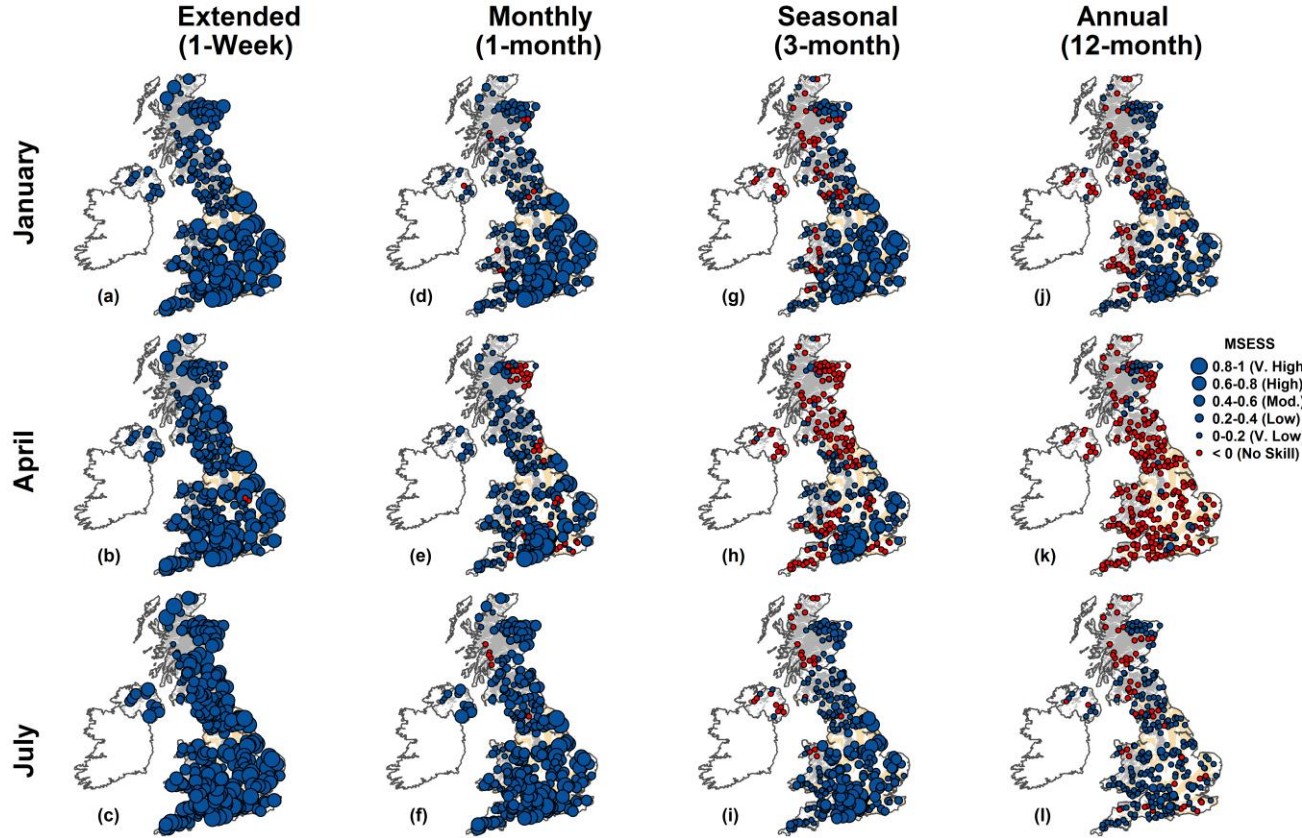

**Figure 7:** ESP skill for individual forecasts made at each of the 314 catchment locations for four sample lead times and three initialisation months. Larger (smaller) circles represent higher (lower) skill from MSESS with blue circles when ESP is more skilful than benchmark climatology and red when ESP has no skill.





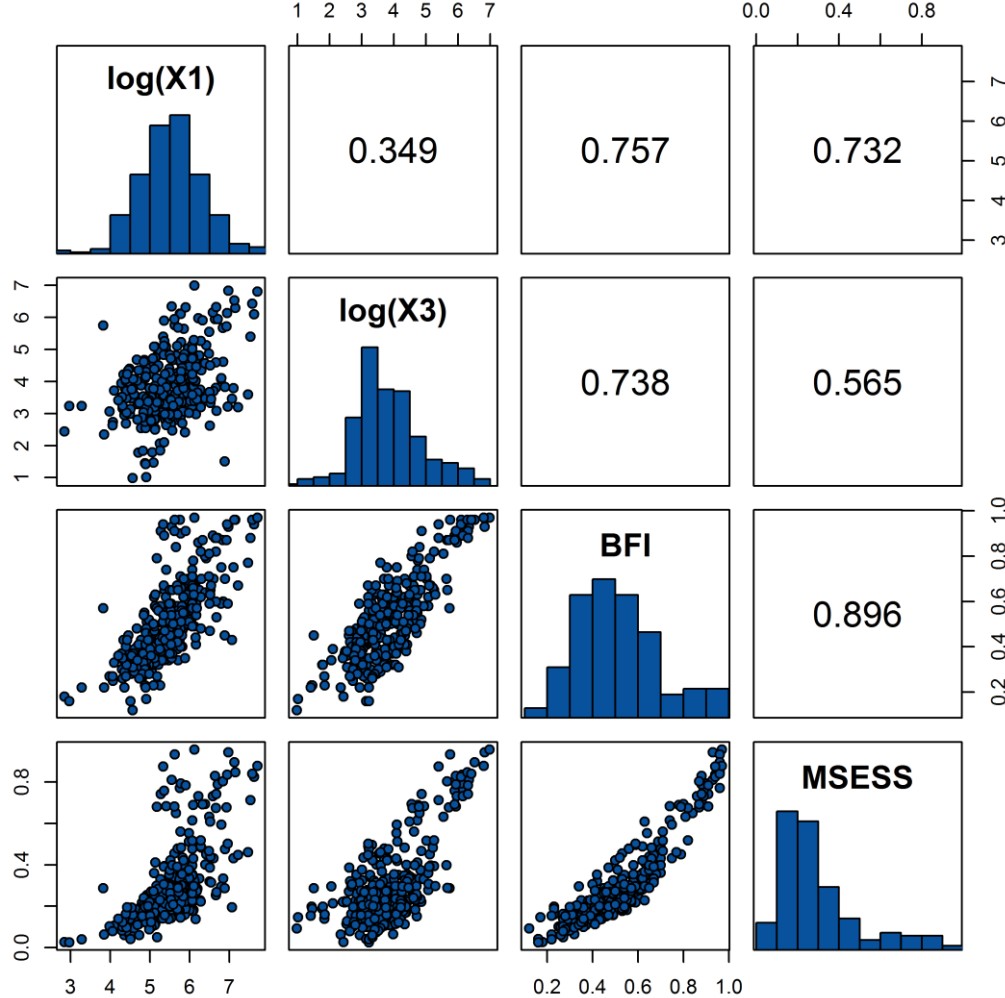

**Figure 8:** Scatterplot matrix between the two calibrated GR4J catchment storage parameters, X1 (soil store capacity [mm]) and X3 (groundwater store capacity [mm]), BFI, and ESP skill (MSESS) with $n$ = 314 using the non-parametric Spearman's rank correlation coefficient $\rho$. Skill is the 1-month MSESS skill score magnitude averaged across all 12 initialisation months. X1 and X3 were re-expressed by taking the log as raw values are heavily positively skewed.