# Peer review of "Benchmarking Ensemble Streamflow Prediction skill in the UK"

_Hydrology and Earth System Sciences, 2017_

## Referee Comment (RC1) · Anonymous Referee #1 · 29 Aug 2017

General Comments:

Overall the paper is well written and makes a positive contribution to the scientific literature within this field. It is well balanced, set out clearly and has a good range of figures. The authors need to address whether they are referring to 'forecasts' or 'projections'. Without conditioning ESP results according to forecast large scale climatic influences i.e. NAO then the results should be termed 'projections' not 'forecasts'. I recommend than with minor revisions the paper should be accepted.

Specific Comments:

1. The paper on many occasions refers to 'ESP forecasts', however as this method is not driven by a meteorological forecast it would be better to refer to these as 'ESP

[Figure]

Projections'.

2. Page 5 lines 11-17: There needs to be greater in depth discussion as to the results presented in Table 2 in the context of other studies. Are the calibration results better than other models/studies?

3. Page 6 Section 3.4: a. Please can the authors clarify what river flow metric are the skill scores being applied to? Is it the skill in comparing the mean daily river flow on a future day 1 day/3day/1 week/2 week etc ahead? Or is it the volume of discharge over the next day/3 days, 1 week/2 weeks,…12 months? b. Did the authors consider using RoC scores to assess skill? Please indicate in the discussion why these were not used.

Technical Corrections:

Page 2 line 10: The Environment Agency implemented operational ESP groundwater level projections in March 2012.

Page 3 line 28: 'NHMP 2017' is the wrong font size

Page 4 line 9: 'hydro climatic regions' – how have these been defined and by whom? please include the reference for their designation.

Page 4 line 13: There are no major sandstone aquifers in Southern England.

Page 4 line 16:' highly productive' – please can you provide an explanation to this term

Page 5 line 7: need to define a UK water year (starting 1st October in year in question)

Page 8 lines 14-15, Page 10 lines 28-29 Page 13 lines 9 and 10: There is generally little variation in monthly rainfall across the year – spring and summer are not necessarily significantly drier. It's the greater evaporative demands in the spring and summer which drives the transition referred to.

Page 11 line 8: The location of the Mole at Kinnersley Manor will not be known by

most readers .It would be better to include the location of all sites mentioned in the text on Figure 1 rather than the insert to Figure 2 which does not include the Mole at Kinnersley Manor.

Figure 1: Include names of sites referred to in the text and Figure 2.

Figure 3: Consider a non linear x axis scale to allow readers to view sub monthly skill results – this is not possible with a linear scale.

Figure 8: axis labels are absent on all x and y axis – is this because they are dimensionless, if not please can these be included on the figure?

---

## Referee Comment (RC2) · G. Thirel (Referee) · 30 Aug 2017

This manuscript presents an evaluation of ESP over the UK. The ensemble forecasts are based on the lumped conceptual GR4J model and past P and PET observations that were resampled as used as input to GR4J. These forecasts are compared to proxy observations (GR4J streamflows using P and PET observations) and a benchmark (resampling of these GR4J streamflows).

This paper is generally well written, very clear, and it makes a significant contribution to the HESS journal. However, I of course have some remarks that would deserve some attention from the authors, some of them not being minor. I am convinced that the authors will be able to handle that efficiently and allow the paper to be published.

[Figure]

Major comments:

The way ESP is thought of in this manuscript is a bit old fashioned in my opinion. It is true that first ESPs were using IHCs and past data, but this is not really the standard nowadays. Indeed, the standard is more what is called in the article NWS ESP. These forecasts are now a well-established method and are the reference, especially up to a month of lead time. I would advise the authors using a more modern terminology in the abstract and article or at least being more specific. Moreover, the justification of the choice of this method should be given.

IHCs influence is high for short lead times and low for large lead times. Following the authors' sentence (P. 8, L. 2-4), that would mean that for short lead times, MSESS and CRPSS should be closer than for long lead times. However, we don't see that on Fig. 4, all lead times seem to have a similar difference between both SSs.

Section 4.1.2: this analysis is interesting. However, there is a second possible entry, in addition to the initialisation month, to take into account in my opinion: the lead time month. Indeed, some periods of the period are easier to predict (typically in between seasons are more prone to changing weather, which is difficult to predict sometimes); that may reflect on the scores, and could explain the differences that are highlighted here. Moreover, some scores can be impacted, for instances, by the streamflow char-acteristics. It is known that Nash-Sutcliffe (not used here) is higher for rivers with strong seasonality, or that CRPS is impacted by the streamflow magnitude (Trinh et al., 2013). I'm wondering to which extent the seasonal analysis (but also the spatial analysis ac-tually!) can be impacted by such issues.

P. 9, L. 21-22: X1 is the production store capacity, and X3 the routing store capacity. It seems difficult to actually link them directly and specifically to soil and groundwater. However, their sum can be considered of the maximum amount of water in the basin (excluding the water in the river and snowpack) and as such it could be of interest including it in Fig. 8.

Section 4.3 aims at finding factors for skill in the model. Did the authors check if the initial states of the model show a correlation with skill? For example, the initial amount of water in the basin, S + R in Fig. 1 of Perrin et al., 2003 (production store + routing store fillings) and the initial snow pack (if a snow model is used) can give good insights (see Singla et al., 2012).

Minor comments:

Abstract: there is a mix between present tense and past tense. Line 14: missing S at ensembleS. Also, lines 21-22 there is a mix between lower, lowest, higher and highest. It is not known from the abstract what the rho symbol represents.

P. 3, L. 21: Section 5 should be Sect. 5 to be consistent with the other occurrences.

P. 3, L. 28: please check all fonts sizes

P. 6, L. 2: initialisation is misspelled

P. 6, L. 3: at p. 5, L. 21, m is the ensemble, not the ensemble size. Also, LT means lead time, it is therefore better not to use LT for designing the number of lead times

P. 6, L. 4: no need for volumes, I think that streamflow is enough

P. 6, L. 15: remove the comma after Wilks

Section 4.1.1, P. 7, L. 26 and later on: do we really need such a precision for all the scores?

P. 9, L. 6: replace "is" with "in" (I think). In this section, percentages sometimes have a space between the figure and the percent sign, sometimes not.

P. 9, L. 13: is "E" actually "SE"?

P. 12, L. 4-6: yes, that definitely has an impact in some basins!

Ghannam et al. reference has some misspelling in the authors' list

[Figure]

Table 1 caption: I would add "R package (Coron et al., 2016, 2017)" after "airGR" and "(Perrin et al., 2003)" at the end of the caption

Table 2 caption: please remind the GR4J calibration period for the parameters that are given here.

Figure 3: I think that "short", "extended", "monthly", "seasonal" and "annual" should be indicating more precisely what they refer to. Maybe use some arrows for this.

References:

Singla, S., Céron, J.P., Martin, E., Regimbeau, F., Déqué, M., Habets, F., Vidal, J.-P. Predictability of soil moisture and river flows over France for the spring season (2012) Hydrology and Earth System Sciences, 16 (1), pp. 201-216.

Trinh, B.N., Thielen-del Pozo, J., Thirel, G. The reduction continuous rank probability score for evaluating discharge forecasts from hydrological ensemble prediction systems (2013) Atmospheric Science Letters, 14 (2), pp. 61-65.

---

## Referee Comment (RC3) · Anonymous Referee #3 · 30 Aug 2017

This paper investigates the performance of the ESP forecast method in the United Kingdom. The authors investigate when, where and why the ESP is skillful, based on a set of 314 catchments and 50 years of hindcasts generated with the GR6J model and data from the UK National River Flow Archive. The forecasts are evaluated with a deterministic and a probabilistic criterion, and compared to modelled streamflow climatology. The authors conclude that the skill decreases exponentially with lead time. Higher skill are observed in forecasts initialized in summer months for lead times up to one month, and in winter and autumn months for seasonal and annual lead times. Higher skill is observed in slow responding catchments with high soil moisture and groundwater reservoirs and less skillful in highly responsive catchments.

General comment

I think that this paper is very well-written and of great quality. The objectives and methods are clearly defined, and therefore easy to read and to follow the scope of the paper. The length of the article and the number of figures were appropriate and the content was always relevant. In addition, this paper fits nicely in the Subseasonal-to-seasonal special issue. This study provides a useful diagnostic of ESP over the UK. I particularly enjoyed how the authors made the link between the spatial and temporal skill patterns and catchment characteristics and seasonal features. I listed some comments and questions below, most of them dealing with methodological aspects, and none of them being major.

Major comments and general questions

In both Twedt et al. (1977) and Day (1985), the abbreviation ESP actually stands for "Extended Streamflow Prediction". It is true that "Ensemble Streamflow Prediction" is widely used, but I think that the original term better conveys the purpose of the method and should be used instead.

P5 L24-25 : "Each of the 51 generated hindcast time-series were then temporally aggregated to provide a forecast of streamflow volume with seamless lead times of 1-day to 12-months, resulting in 365 lead times LT per forecast (leap days were removed)." Do I understand correctly that the streamflow volume for 30 days is obtained by aggregating daily forecasts from day 1 to day 30, and that the streamflow volume for the year aggregates all daily forecasts from day 1 to day 365? If not, could you please clarify? If so, I was confused by the word "lead time" and the analysis involves more factors than just the lead time. Rather than an analysis on lead times, it is an analysis on both aggregation periods and lead times that can be argued to be between 0 days and the last day of the aggregation period. I don't believe this to be real issue, but maybe the authors could be more careful in the way they used the term "lead time". To be more specific, it is the occurrence of "lead times" in Figures 3, 4 and 5 and Section 3.1.1 that triggered this comment.

P5 L28 : Regarding the implementation of the L3OCV method, I was wondering why the authors excluded the subsequent two years but not the preceding two. My guess would be that, operationally, the preceding two years are always available, in any case, while the succeeding two are still missing on the day of the forecast, and adding them will add missing and non-independent information to the calibration-validation procedure. Could the authors say a bit more on that?

P6 L25-27 : "It was found in testing that ESP skill was artificially advantaged (disadvantaged) if cross-validation was not carried out in historic climate forcings (benchmark forecasts), in some cases by ± 15 %." Could you please clarify this sentence?

I was wondering about the authors' choice to use the MSE as deterministic score in this case. If the purpose of the two scores is simply to distinguish between deterministic and probabilistic performances, I would recommend using the Mean Absolute Error (the CRPS value of a deterministic forecast is MAE, Hersbach, 2000) so that, when comparing both scores (e.g. Figure 3), the difference in value is solely due to considering the probabilistic side of the forecast.

Still on the evaluation criteria, given that ESP is a probabilistic ensemble that translates the uncertainty from climatology, I would have liked the authors to focus more on the CRPS than on the MSE, e.g. in Figures 6, 7 and, possibly, 8). Was there a reason to focus on MSE instead?

P7 L17-21 : Is the scale defined for MSESS values or CRPSS values? In the interpretation of Figure 6, it also seems that the threshold value for "Very Low" has shifted to (0, 0.1).

Figure 4 and Table 2: To which extent does the performance of GR4J for each month of the year explains the results obtained for short to medium lead times and presented in Figure 4?

Figure 7: Here, I would have liked to see the maps for November which is cited earlier

in the analysis.

Minor comments

P2 L27 : Please change "out to at a least 7-month lead time" to "out to at least a 7-month lead time"

P3 L28 : "132 catchments that are part the new version" to "132 catchments that are part of the new version"

P6 L2: Please change "initilisation" to "initialisation"

---

## Author Comment (AC1) · 28 Oct 2017

**Harrigan et al. (2017) first response to reviewers - Reviewer 1: RC1 (Anonymous)**

All reviewer 1 comments are labelled consecutively, for example, comment 1 is R#1-1, with our responses to reviewers given in blue text.

General Comments:

R#1-1.   Overall the paper is well written and makes a positive contribution to the scientific literature within this field. It is well balanced, set out clearly and has a good range of figures. The authors need to address whether they are referring to 'forecasts' or 'projections'. Without conditioning ESP results according to forecast large scale climatic influences i.e. NAO then the results should be termed 'projections' not 'forecasts'. I recommend than with minor revisions the paper should be accepted.

We thank the reviewer for their positive and constructive review. We have made the majority of your suggestions and clarify any points raised below. We address your comment about referring to ESP as a forecast below.

Specific Comments:

R#1-2.   1. The paper on many occasions refers to 'ESP forecasts', however as this method is not driven by a meteorological forecast it would be better to refer to these as 'ESP Projections'.

Whilst it is true that ESP does not contain any information about future atmosphere dynamics, it is now standard practice to describe its application in terms of a forecast (e.g., wood et al. (2016), as well as papers within this special issue: e.g., Beckers et al. (2016), Crochemore et al. (2017), and Arnal et al. (2017)). We would like to keep our terminology consistent with these papers but could change it if deemed necessary by the editor.

R#1-3.   2. Page 5 lines 11-17: There needs to be greater in depth discussion as to the results presented in Table 2 in the context of other studies. Are the calibration results better than other models/studies?

The main focus of the paper is not on the hydrological modelling component, it is instead to show that the GR4J model used here could reasonably simulate river flow observations in a wide range of catchments across the UK and could be deemed a viable model for catchment-scale ESP forecasting. The particular focus was on calibration and evaluation of medium range flows (hence why the modified Kling-Gupta efficiency applied to root transformed flows $KGE_{mod}[sqrt]$ was used (i.e. Pg5; L3), and not low (e.g. using log transformed flows) or high flow (e.g. using Nash-Sutcliffe Efficacy (NSE)) metrics, as the method aims to provide ESP forecasts across the full range of the flow regime.

However, we acknowledge that it would be useful to know how our modelling results compare to other models/studies. The most universally used metric for hydrological model calibration/evaluation is the NSE. We have therefore also calculated the NSE for all 314 catchments and will provide a summary of results in **supplementary Figure S1** (see below) and will add individual catchment NSE scores for the calibration and evaluation periods, along with $KGE_{mod}[sqrt]$, in **supplementary Table S1** so that others can make more detailed comparisons.

[Figure]

**Supplementary Figure 1:** Spatial distribution of GR4J model performance for 314 catchments over the calibration (Cal CP [WY1983-2014], top row), and two evaluation periods (Eval P1 [WY1983-1998], middle row and Eval P2 [WY1999-2014], bottom row) for the modified Kling-Gupta efficiency applied to root squared transformed flows (KGEmod[sqrt]) and Nash-Sutcliffe efficiency (NSE) model performance metrics. UK-wide Summary statistics are given in the bottom left for the median and 5th and 95th percentiles.

We propose therefore to insert the following section of text to reflect this review comment on Pg5; L12: "Overall, GR4J performs well against streamflow observations and parameter sets remain stable across P1 and P2 with comparable performance to Crochemore et al. (2017) using GR6J for 16 catchments across France. The NSE was also calculated as it is the most university used metric and so allows comparison to a wider set of studies. Spatial maps and summary statistics of KGE$_{mod}$[sqrt] and NSE are provided in supplementary Figure S1 and, notwithstanding differences in study design, results for GR4J are on par with other large-sample catchment modelling studies in the UK (e.g., Crooks et al. (2009) using the Probability Distributed Model (PDM; Moore, 2007) for 120 catchments)".

R#1-4.  3. Page 6 Section 3.4: a. Please can the authors clarify what river flow metric are the skill scores being applied to? Is it the skill in comparing the mean daily river flow on a future day 1 day/3day/1 week/2 week etc ahead? Or is it the volume of discharge over the next day/3 days, 1 week/2 weeks,…12 months? b. Did the authors consider using RoC scores to assess skill? Please indicate in the discussion why these were not used.

a.)       We thank the reviewer for pointing out that this needs more clarification in the manuscript (which was also queried by 'R#3-3'). The streamflow time-series the evaluation metrics are calculated on is equivalent to the volume of water which flowed from the first (forecast initialisation date) to the last day of the forecast. For simplification, it is expressed in the manuscript in equivalent average daily streamflow (evaluation results are identical for both). We will insert the following text after Pg 5; L25 for clarification: "Note that lead time in this paper refers to the aggregation of mean streamflow over the period from the forecast initialisation date to n days/months ahead in time. So a January ESP forecast with 1-month lead time is the mean streamflow from 1 January to the end of January and a January forecast with 2-month lead time is the mean streamflow from 1 January to end of February".

b.)       The choice of score to evaluate forecast skill is always a difficult subject; in Wilks (2011), the forecast verification chapter on the plethora of available scores/metrics is nearly 100 pages long. The main aim of our work was to investigate the overall performance of the ESP method, and quite rightly pointed out in the 'R#3-7' comment it is an ensemble forecasting method so focus should be on probabilistic scores – we've used one of the most common metrics, the Continuous Ranked Probability Score (CRPS, and skill score) which is a proper score and has the advantage of defaulting to the Mean Absolute Error (MAE) for a deterministic forecast, so is easy to interpret. The ROC diagram and the area under the ROC curve are indeed another way to evaluate the probabilistic forecast performance, but we chose CRPSS for the above reasons.

We have undertaken additional assessment on the use of different forecast evaluation metrics based on suggestions from Reviewer #3 and have taken on board their recommendation to concentrate on the CRPSS instead of the MSESS in the revised manuscript (please see our responses to R#3).

Technical Corrections:

R#1-5.  Page 2 line 10: The Environment Agency implemented operational ESP groundwater level projections in March 2012.

Think will be inserted in Pg2; L13: i.e., "…and also feeds into the Environment Agency's monthly 'Water Situation Reports for England' (operational for groundwater levels in March 2012)".

R#1-6.  Page 3 line 28: 'NHMP 2017' is the wrong font size

Will change in the revised manuscript.

R#1-7.   Page 4 line 9: 'hydro climatic regions' – how have these been defined and by whom? please include the reference for their designation.

The hydroclimatic regions used in the manuscript were defined based on merging contiguous UK hydrometric areas, which are integral river catchments having topographical similarity with outlets to the sea/estuaries (NRFA, 2014), into regions that reflect broad hydrological and climatological patterns in the UK. The approach was based on expert judgment and guided by the Met Office UK regional precipitation regions (HadUKP: https://www.metoffice.gov.uk/hadobs/hadukp/). For example, the division between North-west England & North Wales (NWENW) and South-west England & South Wales (SWESW).

Note that these UK Hydroclimate regions were designated to facilitate the analysis and interpretation of the results, and in particular to investigate if any ESP skill patterns emerged in contrasting hydroclimatic regions. They have, however, no impact on the individual forecast performance. We will edit the revised manuscript on Pg4; L9 for clarity by inserting the following text: "The nine UK Hydroclimate regions were derived by merging contiguous UK hydrometric areas (NRFA, 2014) that reflect broad hydrological and climatological similarity across the UK and are used for aiding interpretation of results".

The UK Hydroclimate Region shapefile, together with metadata, is openly available from the authors or NRFA (nrfa@ceh.ac.uk), and we also highlight this under Sect. 7 – Data availability.

National River Flow Archive: Integrated Hydrological Units of the United Kingdom: Hydrometric Areas with Coastline, NERC Environmental Information Data Centre, Available from: https://doi.org/10.5285/1957166d-7523-44f4-b279-aa5314163237, 2014.

R#1-8.   Page 4 line 13: There are no major sandstone aquifers in Southern England.

We thank the reviewer spotting this. We will remove reference to sandstone.

R#1-9.   Page 4 line 16:' highly productive' – please can you provide an explanation to this term

Highly productive refers to highly permeable aquifers (e.g. Chalk). We agree that this does not fit well here as we are referring to a 'Chalk river', and not specifically the aquifer underneath the catchment so will remove 'highly productive' and change the sentence "in catchments with productive aquifers" in P11; L21 to "in catchments with highly permeable aquifers".

When we refer to a catchment with a large groundwater influence on streamflow, we say the catchment is 'slow responding'.

R#1-10. Page 5 line 7: need to define a UK water year (starting 1st October in year in question)

This was mentioned on Pg4; L3, but we will modify to make it more clear: "Q was retrieved from the NRFA over the longest possible period of observed Q across the 314 stations, 32 water years from 1983 to 2014 (water year from 1 October to 30 September designated by the calendar year in which it ends)".

R#1-11. Page 8 lines 14-15, Page 10 lines 28-29 Page 13 lines 9 and 10: There is generally little variation in monthly rainfall across the year – spring and summer are not necessarily significantly drier. It's the greater evaporative demands in the spring and summer which drives the transition referred to.

We thank the reviewer for this comment as quite rightly the transition between these two half year periods is not significant in terms of precipitation, but the increased evaporative demand. This is summarised better in terms of Soil Moisture Deficits (SMDs). So we will change each of these instances to "April, which in the UK is a transition month between winter months with lowest soil moisture deficits (SMDs) and summer months with highest SMDs".

R#1-12. Page 11 line 8: The location of the Mole at Kinnersley Manor will not be known by most readers .It would be better to include the location of all sites mentioned in the text on Figure 1 rather than the insert to Figure 2 which does not include the Mole at Kinnersley Manor.

This is a good suggestion and we will label the 5 catchments mentioned in Figure 2, along with the Mole at Kinnerley Manor, in Figure 1

R#1-13. Figure 1: Include names of sites referred to in the text and Figure 2.

This will be done as per R#1-12, thanks.

R#1-14. Figure 3: Consider a non linear x axis scale to allow readers to view sub monthly skill results – this is not possible with a linear scale.

We believe the linear scale shows the high rate of skill decay and so would rather keep the linear scale. However, we agree that sub-monthly results are too difficult to see. We have therefore redrawn Figure 3 to include results for short (1- and 3-days) and extended (1- and 2-weeks) lead times (below). Note: this figure is now based only on CRPSS based on R#3 comments on most appropriate choice of skill score.

[Figure]

**Figure 3:** UK-wide mean ESP CRPSS values across all 314 catchments and 12 forecast initialisation months for all 365 lead times (LTs) with short and extended lead times also shown inset for readability. The range of skill scores across catchments at each LT is shown by the semi-transparent 5th and 95th percentile band. Vertical lines represent eight commonly used operational forecasting LTs from short (days) to annual (12-months).

R#1-15. Figure 8: axis labels are absent on all x and y axis – is this because they are dimensionless, if not please can these be included on the figure?

Figure 8 has now been modified based on reviewer comment R#2-5. X1 (mm) and X3 (mm) are now combined are now combined as catchment storage capacity (X1 + X3 in mm) but the log is taken due to it being heavily skewed (as was the case for these variables using in the original manuscript). Therefore the units are 'log mm'. BFI and CRPSS are dimensionless '[-]'. As per your suggestion, axis labels have now been included and will be updated in the revised manuscript accordingly.

Thank you again for your constructive comments,

Shaun.

**References**

Arnal, L., Cloke, H. L., Stephens, E., Wetterhall, F., Prudhomme, C., Neumann, J., Krzeminski, B. and Pappenberger, F.: Skilful seasonal forecasts of streamflow over Europe?, Hydrol. Earth Syst. Sci. Discuss., 2017, 1–27, doi:10.5194/hess-2017-610, 2017.

Beckers, J. V. L., Weerts, A. H., Tijdeman, E. and Welles, E.: ENSO-conditioned weather resampling method for seasonal ensemble streamflow prediction, Hydrol. Earth Syst. Sci., 20(8), 3277–3287, doi:10.5194/hess-20-3277-2016, 2016.

Crochemore, L., Ramos, M.-H., Pappenberger, F. and Perrin, C.: Seasonal streamflow forecasting by conditioning climatology with precipitation indices, Hydrol. Earth Syst. Sci., 21(3), 1573–1591, doi:10.5194/hess-21-1573-2017, 2017.

Crooks, S. M., Kay, A. L. and Reynard, N. S.: Regionalised Impacts of Climate Change on Flood Flows: Hydrological Models, Catchments and Calibration, Centre for Ecology & Hydrology, Environment Agency, Defra, London., 2009.

National River Flow Archive: Integrated Hydrological Units of the United Kingdom: Hydrometric Areas with Coastline, NERC Environmental Information Data Centre, Available from: https://doi.org/10.5285/1957166d-7523-44f4-b279-aa5314163237, 2014.

Wood, A. W., Hopson, T., Newman, A., Brekke, L., Arnold, J. and Clark, M.: Quantifying Streamflow Forecast Skill Elasticity to Initial Condition and Climate Prediction Skill, J. Hydrometeor., 17(2), 651–668, doi:10.1175/JHM-D-14-0213.1, 2016.

---

## Author Comment (AC2) · 28 Oct 2017

All reviewer 2 (Guillaume Thirel') comments are labelled consecutively, for example, comment 1 is R#2-1, with our responses to his comments given in blue text.

R#2-1.  This manuscript presents an evaluation of ESP over the UK. The ensemble forecasts are based on the lumped conceptual GR4J model and past P and PET observations that were resampled as used as input to GR4J. These forecasts are compared to proxy observations (GR4J streamflows using P and PET observations) and a benchmark (resampling of these GR4J streamflows).

This paper is generally well written, very clear, and it makes a significant contribution to the HESS journal. However, I of course have some remarks that would deserve some attention from the authors, some of them not being minor. I am convinced that the authors will be able to handle that efficiently and allow the paper to be published.

We thank Guillaume Thirel very much for his supportive comments and constructive feedback that has helped us refine our paper, particularly his insights on hydrological modelling components.

Major comments:

R#2-2.  The way ESP is thought of in this manuscript is a bit old fashioned in my opinion. It is true that first ESPs were using IHCs and past data, but this is not really the standard nowadays. Indeed, the standard is more what is called in the article NWS ESP. These forecasts are now a well-established method and are the reference, especially up to a month of lead time. I would advise the authors using a more modern terminology in the abstract and article or at least being more specific. Moreover, the justification of the choice of this method should be given.

We fully recognise that ESP, in its traditional form as used here, is a very simple method, and that alternative more-sophisticated ensemble hydrological forecasting techniques are becoming used more and more. We believe, however, there is still a need for benchmarking the skill of such simpler methods as traditional ESP is still considered a good alternative forecasting technique, in the absence of for example expensive seasonal climate forecasts. The choice of evaluating the forecast performance of a simple method like traditional ESP was motivated for three main reasons: 1) to provide a benchmark against which more complex methods could be evaluated for a range of lead times, up to 365 days - this is rarely done (nor possible with more computationally expensive techniques); 2) to identify when/where traditional ESP does not contain sufficient information to generate a skilful hydrological forecast, and henceforth where more complex methods, including use of dynamic atmospheric forecasts, are therefore essential for generating skilful hydrological forecasts; and 3) to formalise the skill of the hydrological seasonal forecasting systems currently used operationally in the UK (within the Hydrological Outlooks UK: http://www.hydoutuk.net/), through a national-scale analysis – the first time this has been done.

We will however edit the revised manuscript to:

 a.) more clearly distinguish that it is ESP in its traditional form we are assessing. For example we will insert the following text in Pg2; L15: "In the traditional formulation of ESP as used in this paper,…" & Pg2 21: "Traditional ESP, while simple, is still widely used today in operational seasonal hydrological forecasting (e.g. US NWS and HOUK) and as a low cost forecast against which to benchmark potential skill improvements from more sophisticated hydro-meteorological ensemble prediction systems".
b.) we will also provide stronger justification why this simple method is still used by many other today and indeed why we are examining it within this manuscript on Pg3; L5: "The previous studies demonstrate that

skilful forecasts can be made using the traditional ESP method at both short and long lead times in many regions around the world and given its relative ease of application and low computational cost remains a valuable ensemble hydrological forecasting approach. Although ESP is being used operationally within the UK, its skill has not yet been investigated at the catchment-scale within a rigorous hindcast experiment and is therefore the focus of this paper".

R#2-3.  IHCs influence is high for short lead times and low for large lead times. Following the authors' sentence (P. 8, L. 2-4) that would mean that for short lead times, MSESS and CRPSS should be closer than for long lead times. However, we don't see that on Fig. 4, all lead times seem to have a similar difference between both SSs.

This comment and the comments from reviewer #3 sparked our curiosity of the impact of using different skill score metrics. We agree with comment R#3-6 that comparing MSESS (as the deterministic measure of ensemble mean) and CRPSS (as the probabilistic measure of full ensemble) in the way we have done in Figure 3 (and on Pg; L2-4 that you are referring to) is misleading as these two scores are not directly comparable. As reviewer #3 points out it is the Mean Absolute Error Skill Score (MAESS) that equals CRPSS for a deterministic forecast (also mentioned in the paper you recommend by Trinh et al., 2013), and would have been better to use instead of MSESS.

We have taken this suggestion on board and have tested four of the most common used metrics for assessing hydrological forecasts: Pearson's correlation coefficient (not a skill score: x = ensemble mean, y = proxy obs), MSESS (deterministic), MAESS (deterministic), and the CRPSS (probabilistic). Results from this analysis show that scores from the MAESS and CRPSS are very similar (see figure S2 below), and that the there is virtually no difference between the skill ensemble mean and full ensemble across lead times or regions (Figure S2 c and d). The results for correlation (Figure S2a) and MSESS (Figure S2b and same as Figure 6 in the original manuscript) are systematically higher than MAESS and CRPSS, not due to IHC influence etc. but simply due to the different formulation of these metrics. Their values on a 0 to 1 scale are not directly comparable. However, it must be made clear that it is only the *magnitude* of values that is different – the results/interpretation of ESP skill remain the same no matter which metric is used (so most/least skilful region, skill across initialisation months etc.).

We have decided to take the advice of review #3 wrt to skill scores and we will concentrate on CRPSS, as ESP is a probabilistic method. Given results are so similar between the full ensemble and deterministic ESP forecasts using MAESS, in the revised manuscript we will only use CRPSS (instead of MSESS) in Figures 3, 4, 5, 6, 7, and 8. Therefore, the text in Pg8; L2-4 referring to your will be modified accordingly. We think it's important to include the results of the comparison of the four scores and will include in as **supplementary Figure S2**.

[Figure]

**Supplementary Figure 2:** Heatmap of mean ESP skill across all 12 forecast initialisation months for the UK and for each of the nine hydroclimate regions ordered from least to most skilful (horizontal axis) at eight sample lead times (vertical axis). Skill is given by the a.) Pearson correlation coefficient (Cor.), b.) Mean Squared Error Skill Score (MSESS), c.) Mean Absolute Error Skill Score (MAESS), and d.) Continuous Ranked Probability Skill Score (CRPSS). Darker (lighter) shades showing higher (lower) skill; individual mean skill values are shown within each cell.

R#2-4. Section 4.1.2: this analysis is interesting. However, there is a second possible entry, in addition to the initialisation month, to take into account in my opinion: the lead time month. Indeed, some periods of the period are easier to predict (typically in between seasons are more prone to changing weather, which is difficult to predict sometimes); that may reflect on the scores, and could explain the differences that are highlighted here. Moreover, some scores can be impacted, for instances, by the streamflow characteristics. It is known that Nash-Sutcliffe (not used here) is higher for rivers with strong seasonality, or that CRPS is impacted by the streamflow magnitude (Trinh et al., 2013). I'm wondering to which extent the seasonal analysis (but also the spatial analysis actually!) can be impacted by such issues.

Thanks for these insights and references. First, the issue with CRPS being impacted by streamflow magnitude (as shown in Trinh et al., 2013) is not a problem in our analysis as we are using the CRPS skill score (CRPSS) so is not dependent on streamflow magnitude. However, the other issues could certainly be playing a minor or major role. As explained in R#2-1 the main aim of this work was to perform the first assessment of ESP skill over a range of lead times at the national scale. In order to identify future possible research avenues, we looked if any simple spatial/temporal patterns emerged from the analysis (i.e. Sections 4.1 and 4.2). The attribution of skill (the 'why' in Section 4.3) is meant as a first assessment of the apparent strong relationship between catchment storage and ESP skill. However, we have a discussion point attribution of different ESP

skills in transition monthly on Pg 10; L30-32:"Factors that might contribute to lower skilled forecasts initialised in spring include potentially higher uncertainty in IHC storage states and larger variability in rainfall across the forecast window (i.e. from late spring to early autumn). Further work should endeavour to attribute the lack of skill during transition seasons but this is outside the scope of this paper".

While we believe a full diagnostic and attribution assessment of the factors responsible for different ESP skills initialised in different times of the year is outside the scope of this paper, as it would require a much more detailed analysis over a complex range of issues, which would lengthen the paper considerably. We will make this much more clear in the revised manuscript and expand the discussion point on Pg10; L30-32 to a wider set of possible explanations of different hydrological forecast performance (e.g. influence of groundwater response, more variable weather conditions over the forecast period, and as GR4J is calibrated using all months there could be some interaction with better/worse simulation of IHCs for different months).

R#2-5.  P. 9, L. 21-22: X1 is the production store capacity, and X3 the routing store capacity. It seems difficult to actually link them directly and specifically to soil and groundwater. However, their sum can be considered of the maximum amount of water in the basin (excluding the water in the river and snowpack) and as such it could be of interest including it in Fig. 8.

We agree that it is very difficult directly link X1 and X3 to soil moisture and groundwater, respectively. However, what is really of interest in this first assessment is the more general question of whether catchment storage is in any way related to ESP performance. We therefore will remove specific reference to linking skill directly to individual soil moisture/groundwater storage capacity model parameter values in the revised manuscript, but instead use your suggestion of viewing (X1 + X3) as total catchment storage capacity (minus snow and water in the river channel). E.g. Research Question 3 on Pg 3; L19 will be changed from "Where is ESP skilful, in terms of individual catchment soil moisture and groundwater storage capacity?" to "Where is ESP skilful, in terms of individual catchment storage capacity?", and will qualify that 'storage capacity is minus snow and water in river channel'.

We have also tested using (X1 + X3) in Section 4.3 and Figure 8, instead of X1 and X2 individually. Results are shown in the below redrawn Figure 8 (left using MSESS and right using the CRPSS, as suggested by reviewer #3). First is that results are virtually the same independent if MSESS or CRPSS is used. Interestingly, the Spearman's correlation coefficient is higher against MSESS for (X1 + X3; $\rho$ = 0.81), than for X1 ($\rho$ = 0.73) or X3 ($\rho$ = 0.57) individually, and is also higher against the BFI for (X1 + X3; $\rho$ = 0.87), than for X1 ($\rho$ = 0.76) or X3 ($\rho$ = 0.74). Therefore, Section 4.3 and Figure 8 will be replaced with the combined catchment storage variable (X1 + X3), instead of X1/X3 individually.

[Figure]

**New Figure 8:** Redrawn using MSESS for comparison with original manuscript (left), and using the CRPSS as is proposed metric within the revised manuscript.

R#2-6.  Section 4.3 aims at finding factors for skill in the model. Did the authors check if the initial states of the model show a correlation with skill? For example, the initial amount of water in the basin, S + R in Fig. 1 of Perrin et al., 2003 (production store + routing store fillings) and the initial snow pack (if a snow model is used) can give good insight (see Singla et al., 2012).

Thank you for this really interesting suggestion. We did not yet explore if initial states show a relationship with skill, but this would certainly be a fruitful avenue for further research into a more detailed attribution of the sources of ESP skill. We feel the revised Figure 8, as outlined in R#2-5, is at a suitable level of detail for the first assessment paper and will certainly pursue this research idea in more detail in our ongoing work, thank you!

Minor comments:

R#2-7.  Abstract: there is a mix between present tense and past tense. Line 14: missing S at ensembleS. Also, lines 21-22 there is a mix between lower, lowest, higher and highest. It is not known from the abstract what the rho symbol represents.

Thank you for these suggestions: We will change issue on P1;L14 to "to produce **a** 51-member ensemble of streamflow hindcasts". We will ensure all tenses are consistent and these text issues are addressed. We will add Spearman's rank correlation coefficient instead of rho symbol.

R#2-8.  P. 3, L. 21: Section 5 should be Sect. 5 to be consistent with the other occurrences.

Will change.

R#2-9.  P. 3, L. 28: please check all fonts sizes

Will change.

R#2-10. P. 6, L. 2: initialisation is misspelled

Will change.

R#2-11. P. 6, L. 3: at p. 5, L. 21, m is the ensemble, not the ensemble size. Also, LT means lead time, it is therefore better not to use LT for designing the number of lead times

Will make consistent.

R#2-12. P. 6, L. 4: no need for volumes, I think that streamflow is enough

Based on review comments from the other two reviewers ow we refer to the aggregation of streamflow has been made more clear as per R#1-4 and R#3-3.

R#2-13. P. 6, L. 15: remove the comma after Wilks

Will change.

R#2-14. Section 4.1.1, P. 7, L. 26 and later on: do we really need such a precision for all the scores?

We agree with the reviewer that the third decimal point in the skill scores/correlations was not necessary and will edit all instances in figures and text throughout the revised manuscript.

R#2-15. P. 9, L. 6: replace "is" with "in" (I think). In this section, percentages sometimes have a space between the figure and the percent sign, sometimes not.

Yes, will change and make spacing consistent with HESS guidelines.

R#2-16. P. 9, L. 13: is "E" actually "SE"?

Yes, good spot, will change.

R#2-17. P. 12, L. 4-6: yes, that definitely has an impact in some basins!

Indeed, while we show that it is only a very small fraction of basins studied has a significant fraction of snow, and even in these cases is usually only for winter months, it is nonetheless an important consideration within ongoing work and this is acknowledged in the text.

R#2-18. Ghannam et al. reference has some misspelling in the authors' list

Will correct.

R#2-19. Table 1 caption: I would add "R package (Coron et al., 2016, 2017)" after "airGR" and "(Perrin et al., 2003)" at the end of the caption

We will also cite these sources in the caption: "* $\overline{F_s}$ calculated using the CemaNeige snow-accounting module (Valéry et al., 2014) within the airGR package (Coron et al., 2016, 2017) applied to the GR4J model (Perrin et al., 2003)".

R#2-20. Table 2 caption: please remind the GR4J calibration period for the parameters that are given here.

The Tabble 2 caption will read in the revised manuscript: "Summary statistics of GR4J calibrated parameters and performance metrics for the UK and nine hydroclimate regions shown in Fig. 1. The median across n catchments within each region is given with the 5$^{th}$ and 95$^{th}$ percentile ranges in brackets. Calibration (Cal) was over the complete period (CP, water years 1983-2014) while evaluation (Eval) for both period 1 (P1, water years 1983-1998) and period 2 (P2, 1999-2014)".

R#2-21. Figure 3: I think that "short", "extended", "monthly", "seasonal" and "annual" should indicating more precisely what they refer to. Maybe use some arrows for this.

These terms refer directly to text on Pg7; L12-13 and Figure 3 has now been redrawn as per comment R#1-14 so we believe it is less cluttered and easier to see the vertical lines these terms directly relate to. We will also make this clearer in the revised figure caption.

References:
Singla, S., Céron, J.P., Martin, E., Regimbeau, F., Déqué, M., Habets, F., Vidal, J.-P. Predictability of soil moisture and river flows over France for the spring season (2012) Hydrology and Earth System Sciences, 16 (1), pp. 201-216.

Trinh, B.N., Thielen-del Pozo, J., Thirel, G. The reduction continuous rank probability score for evaluating discharge forecasts from hydrological ensemble prediction systems
(2013) Atmospheric Science Letters, 14 (2), pp. 61-65.

We thank Guillaume Thirel again for taking the time to provide a constructive and thorough review of our manuscript, his comments will improve our revised manuscript substantially and has given us plenty of ideas for taking the hydrological modelling elements of this work forward.

Kind regards,

Shaun.

---

## Author Comment (AC3) · 28 Oct 2017

All reviewer 3 comments are labelled consecutively, for example, comment 1 is R#3-1, with our responses to reviewers given in blue text.

R#3-1. This paper investigates the performance of the ESP forecast method in the United Kingdom. The authors investigate when, where and why the ESP is skillful, based on a set of 314 catchments and 50 years of hindcasts generated with the GR6J model and data from the UK National River Flow Archive. The forecasts are evaluated with a deterministic and a probabilistic criterion, and compared to modelled streamflow climatology. The authors conclude that the skill decreases exponentially with lead time. Higher skill are observed in forecasts initialized in summer months for lead times up to one month, and in winter and autumn months for seasonal and annual lead times. Higher skill is observed in slow responding catchments with high soil moisture and groundwater reservoirs and less skillful in highly responsive catchments.

General comment

I think that this paper is very well-written and of great quality. The objectives and methods are clearly defined, and therefore easy to read and to follow the scope of the paper. The length of the article and the number of figures were appropriate and the content was always relevant. In addition, this paper fits nicely in the Subseasonal-to-seasonal special issue. This study provides a useful diagnostic of ESP over the UK. I particularly enjoyed how the authors made the link between the spatial and temporal skill patterns and catchment characteristics and seasonal features. I listed some comments and questions below, most of them dealing with methodological aspects, and none of them being major.

We that the reviewer for very supportive comments on our manuscript. The comments and questions around the methodological issues have been assessed and we have decided to take your suggestion about focusing on CRPSS on board throughout the manuscript. We discuss the impact this will have on the revised manuscript below.

Major comments and general questions

R#3-2. In both Twedt et al. (1977) and Day (1985), the abbreviation ESP actually stands for "Extended Streamflow Prediction". It is true that "Ensemble Streamflow Prediction" is widely used, but I think that the original term better conveys the purpose of the method and should be used instead.

We acknowledge the terminology associated with ESP has changed over the years, and recognise that we did not quote appropriately Twedt et al. (1977) and Day (1985). We will edit the text on Pg2; L7-8 to "(Day, 1985; Twedt et al., 1977; originally stood for Extended Streamflow Prediction)".

As per comment from R#1-2, it is now standard practice to describe the traditional ESP approach as 'Ensemble Streamflow Prediction' (e.g., wood et al. (2016), as well as papers within this special issue: e.g., Beckers et al. (2016), Crochemore et al. (2017), and Arnal et al. (2017)). As per R#2-2, we have now made it clearer that we are talking about the 'traditional formulation of ESP' whereby historic meteorological sequences are resampled. We would like to keep our terminology consistent with these papers but could change it if deemed necessary by the editor.

R#3-3. P5 L24-25 : "Each of the 51 generated hindcast time-series were then temporally aggregated to provide a forecast of streamflow volume with seamless lead times of 1-day to 12-months, resulting in 365 lead times LT per forecast (leap days were removed)." Do I understand correctly that the streamflow volume for 30

days is obtained by aggregating daily forecasts from day 1 to day 30, and that the streamflow volume for the year aggregates all daily forecasts from day 1 to day 365? If not, could you please clarify? If so, I was confused by the word "lead time" and the analysis involves more factors than just the lead time. Rather than an analysis on lead times, it is an analysis on both aggregation periods and lead times that can be argued to be between 0 days and the last day of the aggregation period. I don't believe this to be real issue, but maybe the authors could be more careful in the way they used the term "lead time". To be more specific, it is the occurrence of "lead times" in Figures 3, 4 and 5 and Section 3.1.1 that triggered this comment.

We thank the reviewer for pointing out that this needs more clarification in the manuscript (which was also queried by 'R#1-4'). The streamflow time-series the evaluation metrics are calculated on is equivalent to the volume of water which flowed from the first (forecast initialisation date) to the last day of the forecast. For simplification, it is expressed in the manuscript in equivalent average daily streamflow (evaluation results are identical for both). We will insert the following text after Pg 5; L25 for clarification: "Note that lead time in this paper refers to the aggregation of mean streamflow over the period from the forecast initialisation date to n days/months ahead in time. So a January ESP forecast with 1-month lead time is the mean streamflow from 1 January to the end of January and a January forecast with 2-month lead time is the mean streamflow from 1 January to end of February". We now hope this is now clearer.

R#3-4.   P5 L28 : Regarding the implementation of the L3OCV method, I was wondering why the authors excluded the subsequent two years but not the preceding two. My guess would be that, operationally, the preceding two years are always available, in any case, while the succeeding two are still missing on the day of the forecast, and adding them will add missing and non-independent information to the calibration-validation procedure. Could the authors say a bit more on that?

Yes, this is correct. Operationally we have meteorological forcing data to drive ESP up until the forecast initialisation date. In the hindcast experimental design, we will never have exactly the same conditions as the operational case, because we are driving the ESP in the hindcast (e.g. 1965) with precipitation and PET sequences from 'future' periods (e.g. 1967), which clearly we would not have operationally. To make sure the hindcast experiment is as close to operational conditions as practically possible we do not use the current or two succeeding years (i.e. L3OCV), as large-scale climate phenomenon such as the NAO has shown to have multi-season/year persistence in some parts of the UK. We were motivated by an insightful HEPEX blog post by Robertson et al. (2016) which we also cite in the original manuscript: https://hepex.irstea.fr/how-good-is-my-forecasting-method-some-thoughts-on-forecast-evaluation-using-cross-validation-based-on-australian-experiences/.

We will change the section on Pg5; L26-30 to: "Although it is not possible to create a hindcast experiment under exactly the same conditions experienced in operational mode, effort was made to ensure historic climate sequences did not artificially inflate skill (Robertson et al., 2016) by using leave-3-years-out cross-validation (L3OCV) whereby the 12-month forecast window and the two succeeding years were not used as climate forcings. This was done to account for persistence from known large-scale climate-streamflow teleconnections such as the North Atlantic Oscillation with influences lasting from several seasons to years (Dunstone et al., 2016) as this climate information is not available in operational forecasting it should be included either in the hindcast experiment".

R#3-5.   P6 L25-27 : "It was found in testing that ESP skill was artificially advantaged (disadvantaged) if cross-validation was not carried out in historic climate forcings (benchmark forecasts), in some cases by +/-15 %." Could you please clarify this sentence?

This sentence also relates to a point made in the Robertson et al. (2016) HEPEX blog post "Forgetting to cross-validate reference forecasts can unfairly *disadvantage* your forecast method. Remembering to cross-validate the reference forecast (e.g. climatology) is just as important as cross-validating forecasts".

We will replace the text on P6; L25-27 with: "In testing, we performed the evaluation of ESP skill with and without cross-validation of historic climate forcing sequences and climatological benchmark forecasts. It was found that cross-validation was important as in some cases failing to cross-validate historic sequences inflated skill scores (advantaged ESP forecasts) whereas failing to cross-validate climatological benchmark forecasts deflated skill scores (i.e. the benchmark forecast were advantaged thereby disadvantaging ESP forecasts), in some cases by +/-15 %". We hope this is now clearer.

R#3-6. I was wondering about the authors' choice to use the MSE as deterministic score in this case. If the purpose of the two scores is simply to distinguish between deterministic and probabilistic performances, I would recommend using the Mean Absolute Error (the CRPS value of a deterministic forecast is MAE, Hersbach, 2000) so that, when comparing both scores (e.g. Figure 3), the difference in value is solely due to considering the probabilistic side of the forecast.

We thank the reviewer for their recommendation that we have decide to proceed with in the revised manuscript. There is not yet consensus within the hydrological forecasting community on which is the 'best' skill scores to use. We originally decided on MSESS for the deterministic evaluation purely as it has been widely applied and recommended elsewhere. It also has the advantage to being analogous the Nash-Sutcliffe Efficacy (NSE) metric used very widely in hydrological modelling. However, after consideration of your comment and in testing with the MAESS it became clear that the MSESS and CRPSS are not comparable – as you point out for any single ESP you cannot conclude that the ensemble mean (deterministic) is more skilful than the full ensemble (probabilistic) if the MSESS value is higher than the CRPSS value (i.e. Figure 3) – a point we've also responded to R#2-3.

We have taken this suggestion on board and have further tested four of the most common used metrics for assessing hydrological forecasts: Pearson's correlation coefficient (not a skill score: x = ensemble mean, y = proxy obs), MSESS (deterministic), MAESS (deterministic), and the CRPSS (probabilistic). Results from this analysis show that scores from the MAESS and CRPSS are very similar (see figure S2 below), and that the there is virtually no difference between the skill ensemble mean and full ensemble across lead times or regions (Figure S2 c and d). The results for correlation (Figure S2a) and MSESS (Figure S2b and same as Figure 6 in the original manuscript) are systematically higher than MAESS and CRPSS, not due to IHC influence etc. but simply due to the different formulation of these metrics. Their values on a 0 to 1 scale are not directly comparable. However, it must be made clear that it is only the *magnitude* of values that is different – the results/interpretation of ESP skill remain the same no matter which metric is used (so most/least skilful region, skill across initialisation months etc.).

We will concentrate on CRPSS in the revised manuscript, as ESP is a probabilistic method. Given results are so similar between the full ensemble and deterministic ESP forecasts using MAESS, in the revised manuscript we will only use CRPSS (instead of MSESS) in Figures 3, 4, 5, 6, 7, and 8. Therefore, the text in Pg8; L2-4 referring to your will be modified accordingly. We think it's important to include the results of the comparison of the four scores and will include in as **supplementary Figure S2**.

[Figure]

**Supplementary Figure 2:** Heatmap of mean ESP skill across all 12 forecast initialisation months for the UK and for each of the nine hydroclimate regions ordered from least to most skilful (horizontal axis) at eight sample lead times (vertical axis). Skill is given by the a.) Pearson correlation coefficient (Cor.), b.) Mean Squared Error Skill Score (MSESS), c.) Mean Absolute Error Skill Score (MAESS), and d.) Continuous Ranked Probability Skill Score (CRPSS). Darker (lighter) shades showing higher (lower) skill; individual mean skill values are shown within each cell.

R#3-7. Still on the evaluation criteria, given that ESP is a probabilistic ensemble that translates the uncertainty from climatology, I would have liked the authors to focus more on the CRPS than on the MSE, e.g. in Figures 6, 7 and, possibly, 8). Was there a reason to focus on MSE instead?

As per our response to R#3-6, we have now redrawn figures using CRPSS. Below are the redrawn figures 2-8. Basing results on CRPSS does not change the conclusions of the paper in terms of ESP skill, however the reported skills in text using MSESS will be replaced with CRPSS, and the magnitude of the skill is lower. This highlights that the qualitative threshold of what is a 'highly skilful' forecast is strongly metric dependent. For example, the CRPSS for the 6-month January ESP forecast in the Thames is 0.36 with the Pearson correlation coefficient is 0.77 (new Figure 2b below). Sect. 3.4 will be modified to reflect this change. Also, we have revisited Figure 7 and added a new threshold in grey (+/- 0.05) to show the difference between CRPSS values near zero.

[Figure]

**New Figure 2**

[Figure]

**New Figure 3**

[Figure]

**New Figure 4**

[Figure]

**New Figure 5**

[Figure]

**New Figure 6**

[Figure]

**New Figure 7**

[Figure]

**New Figure 8**

R#3-8.  P7 L17-21 : Is the scale defined for MSESS values or CRPSS values? In the interpretation of Figure 6, it also seems that the threshold value for "Very Low" has shifted to (0, 0.1).

Figure 6 does not discuss these qualitative skill categories but rather shows skill per lead time and hydroclimate region having sequential increments at 0.1.

R#3-9.  Figure 4 and Table 2: To which extent does the performance of GR4J for each month of the year explains the results obtained for short to medium lead times and presented in Figure 4?

This is a good point, and was also brought up by R#2-4. We will edit the text as per our response to R#2-4.

R#3-10. Figure 7: Here, I would have liked to see the maps for November which is cited earlier in the analysis.

The new version of Figure 7 is also plotted for November (instead of January in the below figure). As you can see there is very little difference between skill of November and January. The reference to November on Pg8; L13 also includes January. We would therefore prefer to keep Figure 7 with January as these are only sample initialisation months to demonstrate these points.

[Figure]

**New Figure 7** – with November instead of January.

Minor comments

R#3-11. P2 L27 : Please change "out to at a least 7-month lead time" to "out to at least a 7-month lead time"

The study in question did not assess lead times beyond 7-months, so we cannot conclude ESP is not skilful after a 7-month lead time, hence why we used 'out to at least'. We will however make this more clear in the revised manuscript.

R#3-12. P3 L28 : "132 catchments that are part the new version" to "132 catchments that are part of the new version"

Thank you this will be changed. We also note that the number of UK benchmark catchment is 128, not 132. This error will be corrected in the revised manuscript.

R#3-13. P6 L2: Please change "initilisation" to "initialisation"

Will be changed.

We thank the reviewer again for taking the time to give such a positive and considered review of our manuscript. The advice around the best skill scores to use have led us to revise the figures and presentation aspects of the paper and we believe the revised manuscript will be stronger, and more comparable with other papers using ensemble hydrological forecasting.

Kind regards,

Shaun.

---

## Author Response (AR1)

**Harrigan et al. (2017) re-submission of manuscript with final point-by-point response**

Dear QJ,

Thank you for inviting us to revise our paper. We attach i.) our revised abstract, ii.) our revised manuscript, iii.) a final point-by-point response to all reviewers and editor comments with where changes have been made in the revised manuscript, and a marked-up version of the manuscript as required (combined in this pdf), and iv.) revised supplementary tables and figures.

We thank all reviewers again for their time and constructive comments which have helped to greatly improve the quality of the manuscript.

Kind regards,

Shaun.

**Response to editor comment**

The referees have made very thorough and constructive reviews - Thank you!

The authors' responses to the review comments are well considered. Please go ahead and revise the paper.

I note the discussion on the definition of lead times. I tend to follow the convention of the seasonal climate forecasting community, and would encourage you to do the same. Please find the attached slide of mine, which defines target period and lead-time. For example, a forecast issued at the start of January for the target period of January to March has a lead time of zero, while a forecast issued at the start of January for the target period of February to April has a lead time of one month.

Thank you for your slide which very nicely defines forecast target period and lead time. We agree this definition of lead time is indeed common and distinguishing between target period and lead time in this manor has its advantages. However, we would prefer to keep our current definition of lead time which is consistent with what is being used operationally within the UK Hydrological Outlooks (HOUK, Prudhomme et al., 2017), as this paper forms the skill evaluation of one of three of the methods used within HOUK. As per our response to R#1-4 and R#3-3 we made this clearer on Pg6; L11-14 in the revised manuscript: "Following convention in the HOUK, lead time (LT) in this paper refers to the streamflow (expressed as mean daily streamflow) over the period from the forecast initialisation date to n days/months ahead in time. So a January ESP forecast with 1-month lead time is the mean daily streamflow from 1 January to the end of January and a January forecast with 2-month lead time is the mean daily streamflow from 1 January to the end of February". We hope this avoids any confusion, but can revert to your suggestion of zero lead time with n-day/month target period if you feel strongly about this.

**Final response to reviewers**

**Reviewer 1** comments are labelled consecutively, for example, comment 1 is R#1-1, with our responses to reviewers given in blue text.

General Comments:

R#1-1.  Overall the paper is well written and makes a positive contribution to the scientific literature within this field. It is well balanced, set out clearly and has a good range of figures. The authors need to address whether they are referring to 'forecasts' or 'projections'. Without conditioning ESP results according to forecast large scale

climatic influences i.e. NAO then the results should be termed 'projections' not 'forecasts'. I recommend than with minor revisions the paper should be accepted.

We thank the reviewer for their positive and constructive review. We have made the majority of your suggestions and clarify any points raised below. We address your comment about referring to ESP as a forecast below.

Specific Comments:

R#1-2.   1. The paper on many occasions refers to 'ESP forecasts', however as this method is not driven by a meteorological forecast it would be better to refer to these as 'ESP Projections'.

Whilst it is true that ESP does not contain any information about future atmosphere dynamics, it is now standard practice to describe its application in terms of a forecast (e.g., wood et al. (2016), as well as papers within this special issue: e.g., Beckers et al. (2016), Crochemore et al. (2017), and Arnal et al. (2017)). We would like to keep our terminology consistent with these papers but could change it if deemed necessary by the editor.

R#1-3.   2. Page 5 lines 11-17: There needs to be greater in depth discussion as to the results presented in Table 2 in the context of other studies. Are the calibration results better than other models/studies?

The main focus of the paper is not on the hydrological modelling component, it is instead to show that the GR4J model used here could reasonably simulate river flow observations in a wide range of catchments across the UK and could be deemed a viable model for catchment-scale ESP forecasting. The particular focus was on calibration and evaluation of medium range flows metrics (hence why the modified Kling-Gupta efficiency applied to root transformed flows $KGE_{mod}$[sqrt] was used (i.e. Pg5; L3 in the original manuscript), and not low (e.g. using log transformed flows) or high flow (e.g. using Nash-Sutcliffe Efficacy (NSE)), as the hydrological simulation aims to provide ESP forecasts across the full range of the flow regime.

However, we acknowledge that it would be useful to know how our modelling results compare to other models/studies. The most universally used metric for hydrological model calibration/evaluation is the NSE. We have therefore also calculated the NSE for all 314 catchments and provided a summary of results in **supplementary Fig. S1** (see below) and have added individual catchment NSE scores for the calibration and evaluation periods, along with $KGE_{mod}$[sqrt], in **supplementary Table S1** so that others can make more detailed comparisons.

We have also inserted the following text to address this comment on Pg5; L25-31 in the revised manuscript: "Overall, GR4J performs well against streamflow observations and parameter sets remain stable across P1 and P2 with comparable performance to Crochemore et al. (2017) and Poncelet et al. (2017) using GR6J for catchments across France, Germany, and Austria. Overall, GR4J performs well against streamflow observations and parameter sets remain stable across P1 and P2 with comparable performance to Crochemore et al. (2017) and Poncelet et al. (2017) using GR6J for catchments across France, Germany, and Austria. For completeness and comparison with other works, the NSE was calculated as it is the most universally used metric. Spatial maps and summary statistics for KGEmod[sqrt] and NSE are provided in supplementary Fig. S1 and, notwithstanding differences in study design, results for GR4J are on par with other large-sample catchment modelling studies in the UK (e.g. Crooks et al. (2009) using the Probability Distributed Model (PDM; Moore, 2007) for 120 catchments)".

[Figure]

**Supplementary Figure 1:** Spatial distribution of GR4J model performance for 314 catchments over the calibration (Cal CP [WY1983-2014], top row), and two evaluation periods (Eval P1 [WY1983-1998], middle row and Eval P2 [WY1999-2014], bottom row) for the modified Kling-Gupta efficiency applied to root squared transformed flows (KGEmod[sqrt]) and Nash-Sutcliffe efficiency (NSE) model performance metrics. UK-wide Summary statistics are given in the bottom left for the median and 5[th] and 95[th] percentiles.

R#1-4.  3. Page 6 Section 3.4: a. Please can the authors clarify what river flow metric are the skill scores being applied to? Is it the skill in comparing the mean daily river flow on a future day 1 day/3day/1 week/2 week etc ahead? Or is it the volume of discharge over the next day/3 days, 1 week/2 weeks,…12 months? b. Did the authors consider using RoC scores to assess skill? Please indicate in the discussion why these were not used.

a.)     We thank the reviewer for highlighting needs for clarification (also queried by R#3-3). The evaluation metrics are calculated on time series equivalent to the volume of water which flowed from the first day (forecast initialisation date) to the last day of the forecast. For simplification, it is expressed in the manuscript in equivalent average daily streamflow (evaluation results are identical for both). We have inserted the following text in Pg6; L11-14 for clarification: "Following convention in the HOUK, lead time (LT) in this paper refers to the streamflow (expressed as mean daily streamflow) over the period from the forecast initialisation date to $n$ days/months ahead in time. So a January ESP forecast with 1-month lead time is the mean daily streamflow from 1 January to the end of January and a January forecast with 2-month lead time is the mean daily streamflow from 1 January to the end of February".

b.)     The choice of score to evaluate forecast skill is always a difficult subject; in Wilks (2011), the forecast verification chapter on the plethora of available scores/metrics is nearly 100 pages long. The main aim of our work was to investigate the overall performance of the ESP method; as rightly pointed out in R#3-7, ESP is an ensemble forecasting method, so focus should be on probabilistic scores – we've used one of the most common metrics, the Continuous Ranked Probability Score (CRPS, and skill score) which has the advantage of defaulting to the Mean Absolute Error (MAE) for a deterministic forecast, so is easy to interpret. The ROC diagram and the area under the ROC curve are indeed another way to evaluate the probabilistic forecast performance, but we chose CRPSS for the above reasons.

We have undertaken additional assessment on the use of different forecast evaluation metrics based on suggestions from Reviewer #3 and have taken on board their recommendation to concentrate on the CRPSS instead of the MSESS in the revised manuscript (please see our responses to R#3).

Technical Corrections:

R#1-5.  Page 2 line 10: The Environment Agency implemented operational ESP groundwater level projections in March 2012.

This has been inserted in Pg2; L13-14 in the revised manuscript: i.e., "…and also feeds into the Environment Agency's monthly 'Water Situation Report for England' (operational for groundwater levels in March 2012)".

R#1-6.  Page 3 line 28: 'NHMP 2017' is the wrong font size

Changed.

R#1-7.  Page 4 line 9: 'hydro climatic regions' – how have these been defined and by whom? please include the reference for their designation.

The hydroclimatic regions used in the manuscript were defined based on merging contiguous UK hydrometric areas, which are integral river catchments having topographical similarity with outlets to the sea/estuaries (National River Flow Archive, 2014), into regions that reflect broad hydrological and climatological patterns in the UK. The approach was based on expert judgment and guided by the Met Office UK regional precipitation regions (HadUKP: https://www.metoffice.gov.uk/hadobs/hadukp/). For example, the division between North-west England & North Wales (NWENW) and South-west England & South Wales (SWESW).

Note that these UK Hydroclimate Regions were designed to facilitate the analysis and interpretation of the results, and in particular to investigate if any ESP skill patterns emerged in contrasting hydroclimatic regions. They have, however, no impact on the individual forecast performance. We have edited the revised manuscript on Pg4; L19-21 for clarity by inserting the following text: "The nine UK Hydroclimate Regions were derived by merging contiguous UK hydrometric areas (National River Flow Archive, 2014) that reflect broad hydrological and climatological similarity across the UK and are used for aiding interpretation of results".

The UK Hydroclimate Region shapefile, together with metadata, is openly available from the Centre for Ecology & Hydrology (CEH), Wallingford, UK, and we also highlight this under Sect. 7 – Data availability.

R#1-8. Page 4 line 13: There are no major sandstone aquifers in Southern England.

We thank the reviewer spotting this. We have removed reference to sandstone.

R#1-9. Page 4 line 16:' highly productive' – please can you provide an explanation to this term

Highly productive refers to highly permeable aquifers (e.g. Chalk). We agree that this does not fit well here as we are referring to a 'Chalk river', and not specifically the aquifer underneath the catchment so will remove 'highly productive' and change the sentence "in catchments with productive aquifers" in P12; L22 in the revised manuscript to "in catchments with highly permeable aquifers".

When we refer to a catchment with a large groundwater influence on streamflow, we say the catchment is 'slow responding'.

R#1-10. Page 5 line 7: need to define a UK water year (starting 1st October in year in question)

This was mentioned on Pg4; L3, but have modified in the revised manuscript to make clearer (Pg4; L11-12): "Q was retrieved from the NRFA over the longest possible period of observed Q across the 314 stations, 32 water years from 1983 to 2014 (water year from 1 October to 30 September referred to by the calendar year in which it ends)".

R#1-11. Page 8 lines 14-15, Page 10 lines 28-29 Page 13 lines 9 and 10: There is generally little variation in monthly rainfall across the year – spring and summer are not necessarily significantly drier. It's the greater evaporative demands in the spring and summer which drives the transition referred to.

We thank the reviewer for highlighting the need for clarification regarding the transition between these two half year periods being not significant in terms of precipitation but in increased evaporative demand. This is summarised better in terms of Soil Moisture Deficits (SMDs). We have edited the text to "April, which in the UK is a transition month between winter months with lowest soil moisture deficits (SMDs) and summer months with highest SMDs" in the revised manuscript, i.e. Pg9; L10-12 & Pg12-13; L10 & Pg14; L12-13).

R#1-12. Page 11 line 8: The location of the Mole at Kinnersley Manor will not be known by most readers .It would be better to include the location of all sites mentioned in the text on Figure 1 rather than the insert to Figure 2 which does not include the Mole at Kinnersley Manor.

This is a good suggestion and we have labelled the 5 catchments mentioned in Figure 2, along with the Mole at Kinnerley Manor, in Figure 1 in the revised manuscript.

R#1-13. Figure 1: Include names of sites referred to in the text and Figure 2.

This is done as per R#1-12, thanks.

R#1-14. Figure 3: Consider a non linear x axis scale to allow readers to view sub monthly skill results – this is not possible with a linear scale.

We believe the linear scale shows the high rate of skill decay and prefer to keep the linear scale. However, we agree that sub-monthly results are too difficult to see. We have therefore redrawn Figure 3 to include results for short (1- and 3-days) and extended (1- and 2-weeks) lead times. Note: figure 3 in the revised manuscript is now based only on CRPSS based on R#3 comments on most appropriate choice of skill score.

R#1-15. Figure 8: axis labels are absent on all x and y axis – is this because they are dimensionless, if not please can these be included on the figure?

Figure 8 has now been modified based on R#2-5. X1 (mm) and X3 (mm) are now combined as catchment storage capacity (X1 + X3 in mm) but log transformed (using the natural log) because of the large skew in the values (as was done in the original manuscript). Therefore the units are 'log mm'. BFI and CRPSS are dimensionless '[-]'. Axis labels have now been included.

R#2-1.  This manuscript presents an evaluation of ESP over the UK. The ensemble forecasts are based on the lumped conceptual GR4J model and past P and PET observations that were resampled as used as input to GR4J. These forecasts are compared to proxy observations (GR4J streamflows using P and PET observations) and a benchmark (resampling of these GR4J streamflows).

This paper is generally well written, very clear, and it makes a significant contribution to the HESS journal. However, I of course have some remarks that would deserve some attention from the authors, some of them not being minor. I am convinced that the authors will be able to handle that efficiently and allow the paper to be published.

We thank Guillaume Thirel very much for his supportive comments and constructive feedback that has helped us refine our paper, particularly his insights on hydrological modelling components.

Major comments:

R#2-2.  The way ESP is thought of in this manuscript is a bit old fashioned in my opinion. It is true that first ESPs were using IHCs and past data, but this is not really the standard nowadays. Indeed, the standard is more what is called in the article NWS ESP. These forecasts are now a well-established method and are the reference, especially up to a month of lead time. I would advise the authors using a more modern terminology in the abstract and article or at least being more specific. Moreover, the justification of the choice of this method should be given.

We fully recognise that ESP, in its traditional form as used here, is a very simple method, and that alternative more sophisticated ensemble hydrological forecasting techniques are becoming increasingly used. We believe, however, there is still a need for benchmarking the skill of such simpler methods, as traditional ESP is still considered a good alternative forecasting technique, in the absence of for example expensive seasonal climate forecasts. The choice of evaluating the forecast performance of a simple method like traditional ESP was motivated for three main reasons: 1) to provide a benchmark against which more complex methods could be evaluated for a range of lead times, up to 365 days - this is rarely done (nor possible with more computationally expensive techniques); 2) to identify when/where traditional ESP does not contain sufficient information to generate a skilful hydrological forecast, and henceforth where more complex methods, including use of dynamic atmospheric forecasts, are therefore essential for generating skilful hydrological forecasts; and 3) to formalise the skill of the hydrological seasonal forecasting systems currently used operationally in the UK (within the Hydrological Outlooks UK: http://www.hydoutuk.net/), through a national-scale analysis – the first time this has been done.

We have however edited the revised manuscript to:

a.)  More clearly distinguish that it is ESP in its traditional form we are assessing: Pg2; L16: "In the traditional formulation of ESP as used in this paper,…" & Pg2 23-25: "Traditional ESP, while simple, is still widely used today in operational seasonal hydrological forecasting (e.g. US NWS and HOUK) and as a low cost forecast against which to benchmark potential skill improvements from more sophisticated hydro-meteorological ensemble prediction systems".

b.)  Give a stronger justification why the simple ESP method is still used by many others today and indeed why we are examining it within this manuscript on Pg3; L7-11: "The previous studies demonstrate that the traditional ESP method is skilful at both short and long lead times in many regions around the world and given its relative ease of application and low computational cost remains a valuable ensemble hydrological forecasting approach. Although ESP is being used operationally within the UK, its skill has not yet been investigated at the catchment-scale within a rigorous hindcast experiment and is therefore the focus of this paper".

R#2-3. IHCs influence is high for short lead times and low for large lead times. Following the authors' sentence (P. 8, L. 2-4) that would mean that for short lead times, MSESS and CRPSS should be closer than for long lead times. However, we don't see that on Fig. 4, all lead times seem to have a similar difference between both SSs.

This comment and the comments from reviewer #3 sparked our curiosity of the impact of using different skill score metrics. We agree with R#3-6 that comparing MSESS (as the deterministic measure of ensemble mean) and CRPSS (as the probabilistic measure of full ensemble) as originally done in Figure 3, 4, and 5 (and on Pg8; L2-4 in the original manuscript that you are referring to) is misleading as these two scores are not directly comparable. As reviewer #3 points out it is the Mean Absolute Error Skill Score (MAESS) that equals CRPSS for a deterministic forecast (also mentioned in Trinh et al. (2013) as recommend by you), and would have been better to use instead of MSESS. We have therefore changed the analysis to replace MSESS by MAESS, and in fact see virtually the same results for probabilistic (using CRPSS) and deterministic (using MAESS), and as a consequence this section of text has been removed in the revised manuscript. The following text has been added to the revised manuscript on Pg8; L26-27 instead: "Skill scores for the deterministic ESP ensemble mean (measured by MAESS) are virtually the same as those for probabilistic forecasts (measured by CRPSS) for all lead times and regions (see Fig. S2c and d)".

A more detailed response is given in R#3-6 below (relevant here but not repeated for brevity) justifying changing the core analysis to be based on CRPSS instead of MSESS.

R#2-4. Section 4.1.2: this analysis is interesting. However, there is a second possible entry, in addition to the initialisation month, to take into account in my opinion: the lead time month. Indeed, some periods of the period are easier to predict (typically in between seasons are more prone to changing weather, which is difficult to predict sometimes); that may reflect on the scores, and could explain the differences that are highlighted here. Moreover, some scores can be impacted, for instances, by the streamflow characteristics. It is known that Nash-Sutcliffe (not used here) is higher for rivers with strong seasonality, or that CRPS is impacted by the streamflow magnitude (Trinh et al., 2013). I'm wondering to which extent the seasonal analysis (but also the spatial analysis actually!) can be impacted by such issues.

Thanks for these insights and references. First, the issue with CRPS being impacted by streamflow magnitude (as shown in Trinh et al., 2013) is not relevant in our analysis as we are using the CRPS skill score (CRPSS), independent on streamflow magnitude. However, the other issues highlighted could certainly be playing a minor or major role. As explained in our response to R#2-1 the main aim of this work was to perform the first assessment of ESP skill over a range of lead times at the national scale. In order to identify future possible research avenues, we looked if any simple spatial/temporal patterns emerged from the analysis (i.e. Sections 4.1 and 4.2). The attribution of skill (the 'why' in Section 4.3) is meant as a first assessment of the apparent strong relationship between catchment storage and ESP skill.

While we believe a full diagnostic and attribution assessment of the factors responsible for different ESP skills initialised in different times of the year is outside the scope of this paper, as it would require a much more detailed analysis over a complex range of issues, which would lengthen the paper considerably. We have however added a discussion point on the matter and modified the text on Pg 11; L28-32 in the revised manuscript to: "Factors that might contribute to lower skilled forecasts initialised in spring, and indeed to differences in skill across all initialisation months, include: potentially higher variability in IHC storage states, changing variability in rainfall across the forecast window (e.g. from late spring to early autumn), and differences in model performance for different months over the year due to the global calibration of GR4J. Given the answer is likely a combination of many of these factors, among others, further work should endeavour to attribute differences in skill during different times of the year but this is outside the scope of this paper".

R#2-5.  P. 9, L. 21-22: X1 is the production store capacity, and X3 the routing store capacity. It seems difficult to actually link them directly and specifically to soil and groundwater. However, their sum can be considered of the maximum amount of water in the basin (excluding the water in the river and snowpack) and as such it could be of interest including it in Fig. 8.

We agree that it is very difficult directly link X1 and X3 to soil moisture and groundwater, respectively. However, what is really of interest in this first assessment is the more general question of whether catchment storage is in any way related to ESP performance. We have therefore removed specific reference to linking skill directly to individual soil moisture/groundwater storage capacity model parameter values in the revised manuscript, but instead use your suggestion (thank you!). The text on Pg10; L19-23 in the revised manuscript now reads: "It is difficult to link X1 and X3 specifically to soil moisture and groundwater storage capacity, respectively, as GR4J is not a physically-based hydrological model. However, their sum (X1 + X3) can be considered an estimate of total catchment storage (excluding water in the river channel and snowpack). Total catchment storage (X1 + X3) is strongly positively (non-linearly) correlated with BFI ($\rho = 0.87$); catchments with high BFIs tend to have much higher than average catchment storage capacity".

We now use total catchment storage capacity (X1 + X3) in Section 4.3 and Figure 8, instead of X1 and X2 individually. Results are shown in the below redrawn Figure 8 (left using MSESS and right using the CRPSS, as suggested by reviewer #3). First is that results are virtually the same independent if MSESS or CRPSS is used. Interestingly, the Spearman's correlation coefficient is higher against MSESS for (X1 + X3) ($\rho = 0.81$), than for X1 ($\rho = 0.73$) or X3 ($\rho = 0.57$) individually, and is also higher against the BFI for (X1 + X3) ($\rho = 0.87$), than for X1 ($\rho = 0.76$) or X3 ($\rho = 0.74$). Therefore, Section 4.3 and Figure 8 has been replaced with the combined catchment storage variable (X1 + X3), instead of X1/X3 individually, and for CRPSS rather than MSESS.

[Figure]

**New Figure 8:** Redrawn using MSESS for comparison with original manuscript (left), and using the CRPSS as is proposed metric within the revised manuscript.

R#2-6.  Section 4.3 aims at finding factors for skill in the model. Did the authors check if the initial states of the model show a correlation with skill? For example, the initial amount of water in the basin, S + R in Fig. 1 of Perrin et al., 2003 (production store + routing store fillings) and the initial snow pack (if a snow model is used) can give good insight (see Singla et al., 2012).

Thank you for this really interesting suggestion. We did not yet explore if initial states show a relationship with skill, but this would certainly be a fruitful avenue for further research into a more detailed attribution of the sources of ESP skill. We feel the revised Figure 8, as outlined in R#2-5, is at a suitable level of detail for the first assessment paper and will certainly pursue this research idea in more detail in our ongoing work, thank you!

Minor comments:

R#2-7. Abstract: there is a mix between present tense and past tense. Line 14: missing S at ensembleS. Also, lines 21-22 there is a mix between lower, lowest, higher and highest. It is not known from the abstract what the rho symbol represents.

Thank you for these suggestions: We have changed this to: "to produce **a** 51-member ensemble of streamflow hindcasts", we have also revised the tenses and now spell out the rho symbol as "Spearman's rank correlation coefficient".

R#2-8. P. 3, L. 21: Section 5 should be Sect. 5 to be consistent with the other occurrences.

Changed.

R#2-9. P. 3, L. 28: please check all fonts sizes

Changed.

R#2-10. P. 6, L. 2: initialisation is misspelled

Changed.

R#2-11. P. 6, L. 3: at p. 5, L. 21, m is the ensemble, not the ensemble size. Also, LT means lead time, it is therefore better not to use LT for designing the number of lead times

We now do not refer to m or LT in this way as per your suggestion.

R#2-12. P. 6, L. 4: no need for volumes, I think that streamflow is enough

Volumes are now not referred to as per your suggestion.

R#2-13. P. 6, L. 15: remove the comma after Wilks

Changed.

R#2-14. Section 4.1.1, P. 7, L. 26 and later on: do we really need such a precision for all the scores?

We agree with the reviewer that the third decimal point in the skill scores/correlations was not necessary and have changed all instances in figures and text throughout the revised manuscript.

R#2-15. P. 9, L. 6: replace "is" with "in" (I think). In this section, percentages sometimes have a space between the figure and the percent sign, sometimes not.

Yes, have changed and made spacing consistent throughout.

R#2-16. P. 9, L. 13: is "E" actually "SE"?

Yes, good spot, changed.

R#2-17. P. 12, L. 4-6: yes, that definitely has an impact in some basins!

Indeed, while we show that it is only a very small fraction of basins studied that have a significant fraction of snow, and usually only for winter months, it is nonetheless an important consideration within ongoing work and this is acknowledged in the text.

R#2-18. Ghannam et al. reference has some misspelling in the authors' list

Changed.

R#2-19. Table 1 caption: I would add "R package (Coron et al., 2016, 2017)" after "airGR" and "(Perrin et al., 2003)" at the end of the caption

Have now also cited these sources in the caption: "* $\overline{F_s}$ calculated using the CemaNeige snow-accounting module (Valéry et al., 2014) within the airGR package (Coron et al., 2016, 2017) applied to the GR4J model (Perrin et al., 2003)".

R#2-20. Table 2 caption: please remind the GR4J calibration period for the parameters that are given here.

The Table 2 caption now reads in the revised manuscript: "Summary statistics of GR4J calibrated parameters and performance metrics for the UK and nine hydroclimate regions shown in Fig. 1. The median across n catchments within each region is given with the 5th and 95th percentile ranges in brackets. Calibration (Cal) was over the complete period (CP, water years 1983-2014) while evaluation (Eval) for both period 1 (P1, water years 1983-1998) and period 2 (P2, 1999-2014)".

R#2-21. Figure 3: I think that "short", "extended", "monthly", "seasonal" and "annual" should indicating more precisely what they refer to. Maybe use some arrows for this.

These terms refer directly to text on Pg7; L12-13 in the original manuscript and Figure 3 has now been redrawn as per R#1-14 in the revised manuscript so we believe it is less cluttered and easier to see the vertical lines these terms directly relate to. This is also clearer in the revised figure 3 caption.

R#3-1.  This paper investigates the performance of the ESP forecast method in the United Kingdom. The authors investigate when, where and why the ESP is skillful, based on a set of 314 catchments and 50 years of hindcasts generated with the GR6J model and data from the UK National River Flow Archive. The forecasts are evaluated with a deterministic and a probabilistic criterion, and compared to modelled streamflow climatology. The authors conclude that the skill decreases exponentially with lead time. Higher skill are observed in forecasts initialized in summer months for lead times up to one month, and in winter and autumn months for seasonal and annual lead times. Higher skill is observed in slow responding catchments with high soil moisture and groundwater reservoirs and less skillful in highly responsive catchments.

General comment

I think that this paper is very well-written and of great quality. The objectives and methods are clearly defined, and therefore easy to read and to follow the scope of the paper. The length of the article and the number of figures were appropriate and the content was always relevant. In addition, this paper fits nicely in the Subseasonal-to-seasonal special issue. This study provides a useful diagnostic of ESP over the UK. I particularly enjoyed how the authors made the link between the spatial and temporal skill patterns and catchment characteristics and seasonal features. I listed some comments and questions below, most of them dealing with methodological aspects, and none of them being major.

We thank the reviewer for very supportive comments on our manuscript. The comments and questions around the methodological issues have been assessed and we have decided to take on board your suggestion about focusing on CRPSS and so have changed the figures and text throughout the revised manuscript. We discuss the impact this has had on the revised manuscript below.

Major comments and general questions

R#3-2.  In both Twedt et al. (1977) and Day (1985), the abbreviation ESP actually stands for "Extended Streamflow Prediction". It is true that "Ensemble Streamflow Prediction" is widely used, but I think that the original term better conveys the purpose of the method and should be used instead.

We acknowledge the terminology associated with ESP has changed over the years, and recognise that we did not quote appropriately Twedt et al. (1977) and Day (1985). We have edited the text on Pg2; L7-8 to "(Day, 1985; Twedt et al., 1977; originally stood for Extended Streamflow Prediction)".

As per R#1-2, it is now common practice to describe the traditional ESP approach as 'Ensemble Streamflow Prediction' (see response). As per R#2-2, we have now made it clearer that we are talking about the 'traditional formulation of ESP' whereby historic meteorological sequences are resampled. We would like to keep our terminology consistent with these papers but could change it if deemed necessary by the editor.

R#3-3.  P5 L24-25 : "Each of the 51 generated hindcast time-series were then temporally aggregated to provide a forecast of streamflow volume with seamless lead times of 1-day to 12-months, resulting in 365 lead times LT per forecast (leap days were removed)." Do I understand correctly that the streamflow volume for 30 days is obtained by aggregating daily forecasts from day 1 to day 30, and that the streamflow volume for the year aggregates all daily forecasts from day 1 to day 365? If not, could you please clarify? If so, I was confused by the word "lead time" and the analysis involves more factors than just the lead time. Rather than an analysis on lead times, it is an analysis on both aggregation periods and lead times that can be argued to be between 0 days and the last day of the aggregation period. I don't believe this to be real issue, but maybe the authors could be more careful in the way they used the term "lead time". To be more specific, it is the occurrence of "lead times" in Figures 3, 4 and 5 and Section 3.1.1 that triggered this comment.

We thank the reviewer for pointing out that this needs more clarification in the manuscript, and answered in R#1-4a (not repeated here for brevity).

R#3-4.  P5 L28 : Regarding the implementation of the L3OCV method, I was wondering why the authors excluded the subsequent two years but not the preceding two. My guess would be that, operationally, the preceding two years are always available, in any case, while the succeeding two are still missing on the day of the forecast, and adding them will add missing and non-independent information to the calibration-validation procedure. Could the authors say a bit more on that?

Yes, this is correct. Operationally we have meteorological forcing data to drive ESP up until the forecast initialisation date. In the hindcast experimental design, we will never have exactly the same conditions as the operational case, because we are driving the ESP in the hindcast (e.g. 1965) with precipitation and PET sequences from 'future' periods (e.g. 1967), which clearly we would not have operationally. To make sure the hindcast experiment is as close to operational conditions as practically possible we do not use the current or two succeeding years (i.e. L3OCV), as large-scale climate phenomenon such as the NAO has shown to have multi-season/year persistence in some parts of the UK. We were motivated by an insightful HEPEX blog post by Robertson et al. (2016) which we also cited in the original manuscript: https://hepex.irstea.fr/how-good-is-my-forecasting-method-some-thoughts-on-forecast-evaluation-using-cross-validation-based-on-australian-experiences/.

We have modified this section of text (now at Pg6; L15-21) to: "Although it is not possible to create a hindcast experiment under exactly the same conditions experienced in operational mode, effort was made to ensure historic climate sequences did not artificially inflate skill (see Robertson et al., 2016) by using leave-3-years-out cross-validation (L3OCV) whereby the 12-month forecast window and the two succeeding years were not used as climate forcings. This was done to account for persistence from known large-scale climate-streamflow teleconnections such as the North Atlantic Oscillation with influences lasting from several seasons to years (Dunstone et al., 2016). Because this climate information could be an advantage, but is not available in operational forecasting, it was not used in the hindcast experiment.

R#3-5.  P6 L25-27 : "It was found in testing that ESP skill was artificially advantaged (disadvantaged) if cross-validation was not carried out in historic climate forcings (benchmark forecasts), in some cases by +/-15 %." Could you please clarify this sentence?

This sentence also relates to a point made in the Robertson et al. (2016) HEPEX blog post "Forgetting to cross-validate reference forecasts can unfairly *disadvantage* your forecast method. Remembering to cross-validate the reference forecast (e.g. streamflow climatology used here) is just as important as cross-validating ESP forecasts".

We have replaced the text on P7; L15-19 with: "In testing, we performed the skill evaluation with and without cross-validation of ESP forecasts and streamflow climatology benchmark forecasts. It was found that cross-validation was important as in some cases failing to cross-validate ESP forecasts inflated skill scores whereas failing to cross-validate climatological benchmark forecasts deflated skill scores (i.e. the benchmark forecast was advantaged thereby disadvantaging ESP forecasts), in some cases skill scores were advantaged/disadvantaged by +/-15 %".

R#3-6.  I was wondering about the authors' choice to use the MSE as deterministic score in this case. If the purpose of the two scores is simply to distinguish between deterministic and probabilistic performances, I would recommend using the Mean Absolute Error (the CRPS value of a deterministic forecast is MAE, Hersbach, 2000) so that, when comparing both scores (e.g. Figure 3), the difference in value is solely due to considering the probabilistic side of the forecast.

We thank the reviewer for their recommendation and have implemented the suggestion in the revised manuscript. There is not yet consensus within the hydrological forecasting community on which is the 'best' skill score/combination of scores to use. We originally decided on MSESS for the deterministic evaluation purely as it has been widely applied and recommended elsewhere. It also has the advantage to being analogous the Nash-Sutcliffe Efficacy (NSE) metric used very widely in hydrological modelling. However, after consideration of your comment and in testing with the MAESS it became clear that the way MSESS and CRPSS were represented in Figures 3, 4, and 5 could be confusing as they are not directly comparable – as you point out for any single ESP you cannot conclude that the ensemble mean (deterministic) is more skilful than the full ensemble (probabilistic) if the MSESS value is higher than the CRPSS value – a point responded to R#2-3.

We have further tested four of the most common used metrics for assessing hydrological forecasts: Pearson's correlation coefficient (not a skill score: x = ensemble mean, y = proxy obs), MSESS (deterministic), MAESS (deterministic), and the CRPSS (probabilistic). Results from this analysis show that scores from the MAESS and CRPSS are very similar (see the new **supplementary figure S2 below**), and that there is virtually no difference between the skill ensemble mean and full ensemble across lead times or regions (Figure S2 c and d). The results for correlation (Figure S2a) and MSESS (Figure S2b and same as Figure 6 in the original manuscript) are systematically higher than MAESS and CRPSS, not due to IHC influence etc. but simply due to the different formulation of these metrics. Their values on a 0 to 1 scale are not directly comparable. However, it must be made clear that it is only the *magnitude* of values that is different – the results/interpretation of ESP skill remain largely the same no matter which metric is used (so most/least skilful region, skill across initialisation months etc.).

We have now concentrated on CRPSS (instead of MSESS originally) in the revised manuscript, as ESP is a probabilistic method. Given results are so similar between the full ensemble (CRPSS) and deterministic ESP forecasts using MAESS, in the revised manuscript we have only used CRPSS in Figures 3, 4, 5, 6, 7, and 8. Therefore, we have added the following text on Pg 7; L26- Pg 8; L2: "The CRPS is one of the most recommended scores for evaluation of overall hydrological ensemble forecast performance (Pappenberger et al., 2015). However, several commonly used metrics were also calculated for evaluation of deterministic ESP performance (from the ESP ensemble mean): Pearson correlation coefficient (Cor.), the mean squared error skill score (MSESS), and the deterministic equivalent to CRPSS, the mean absolute error skill score (MAESS). The pattern of results in terms of where and when ESP is most/least skilful was found to be independent of chosen metric, with virtually identical results between probabilistic (using CRPSS) and deterministic (using MAESS) results (see supplementary Fig. S2), and so for brevity the remainder of paper is based on CRPSS only".

[Figure]

[Figure]

**Supplementary Figure 2:** Mean ESP skill across all 12 forecast initialisation months for the UK and for each of the nine hydroclimate regions ordered from least to most skilful (horizontal axis) at eight sample lead times (vertical axis). Skill is given by the a.) Pearson correlation coefficient (Cor.), b.) Mean Squared Error Skill Score (MSESS), c.) Mean Absolute Error Skill Score (MAESS), and d.) Continuous Ranked Probability Skill Score (CRPSS). Darker (lighter) shades showing higher (lower) skill; individual mean skill values are shown within each cell.

R#3-7.   Still on the evaluation criteria, given that ESP is a probabilistic ensemble that translates the uncertainty from climatology, I would have liked the authors to focus more on the CRPS than on the MSE, e.g. in Figures 6, 7 and, possibly, 8). Was there a reason to focus on MSE instead?

As per our response to R#3-6 above, we have now redrawn figures 3-8 using CRPSS but this do not change the conclusions of the paper in terms of ESP skill. Note that the now reported skill magnitudes using CRPSS are lower than previous MSESS. This highlights that the qualitative threshold of what is a 'highly skilful' forecast is strongly metric dependent. For example, the CRPSS for the 6-month January ESP forecast in the Thames is 0.36 with the Pearson correlation coefficient is 0.77 (new Figure 2b in the revised manuscript). Sect. 3.4 has been modified to reflect this change. Also, we have revisited Figure 7 and added a new threshold in grey (between +/- 0.05) called 'neutrally skilful' after Bennett et al. (2017) to show the difference between CRPSS values near zero. The text on qualitative thresholds has been modified on Pg 8; L13-18 in line with the above changes:

"Reducing accuracy of a forecast to a numeric skill metric value is abstract and difficult to interpret. Throughout the results and discussion sections skill score values are assigned qualitative descriptions according to degree of skill based on the CRPSS: Very High [0.75, 1]; High [0.5, 0.75); Moderate [0.25, 0.5); Low (0, 0.25); No Skill = 0, and Negative Skill < 0; CRPSS values which are near zero, defined between ± 0.05, are regarded as 'neutrally skilful' (after Bennett et al., 2017). Five example 1965-2015 hindcast time-series with skills ranging from very high to negative skill are visualised in Fig. 2 and act as a graphical reference in the remainder of the paper to aid interpretation of skill".

R#3-8.   P7 L17-21 : Is the scale defined for MSESS values or CRPSS values? In the interpretation of Figure 6, it also seems that the threshold value for "Very Low" has shifted to (0, 0.1).

Figure 6 does not discuss these qualitative skill categories but rather shows skill per lead time and hydroclimate region having sequential increments at 0.1.

R#3-9.   Figure 4 and Table 2: To which extent does the performance of GR4J for each month of the year explains the results obtained for short to medium lead times and presented in Figure 4?

This is a good point, also brought up by R#2-4. This has been revised - see response to R#2-4 - to include reference here to the potential performance of GR4J throughout the year. This is an interesting point but is outside the scope of the paper.

R#3-10. Figure 7: Here, I would have liked to see the maps for November which is cited earlier in the analysis.

The aim of Figure 7 is to demonstrate the value of mapping skill scores at the individual catchment scale to highlight the high degree of sub-region heterogeneity. To do this we needed to select a sample of lead times (here, four: 1-week, 1-month, 3-month, and 12-month) and a sample initialisation months (here, January, April, and July in the original manuscript). The choice of three interesting initialisation months was mainly guided by results from Sect. 4.1.2. January as being an interesting month representative of months when soil moisture deficits (SMDs) are low, April representative of spring SMD transition conditions, where ESP skills have shown to be lowest in the UK, and August which is now the most skilful month, on average, for lead times up to 1-month using the CRPSS (was July using MSESS in the original manuscript). We could add another initialisation month (e.g. November as you suggested) but there is little additional information and results for January are largely representative, see the below figure for November also. We would prefer to keep just three initialisation months for simplicity and to save space, but could change Figure 7 to the below if deemed preferable by the editor.

Both versions of the Figures are below: three initialisation months (January, April, and August) and four initialisation months are one per mid-season (i.e. January, April, July, and November).

**Figure 7** – 3 x 4 (as in the revised manuscript):

[Figure]

**Figure 7** – 4 x 4 (could change to this version if necessary):

[Figure]

Minor comments

R#3-11.P2 L27 : Please change "out to at a least 7-month lead time" to "out to at least a 7-month lead time"

Modified, thanks.

R#3-12. P3 L28 : "132 catchments that are part the new version" to "132 catchments that are part of the new version"

Thank you this has been changed. We also note that the number of UK benchmark catchment is 128, not 132. This error has been corrected in the revised manuscript.

R#3-13. P6 L2: Please change "initilisation" to "initialisation"

Changed.

**References**

[revised manuscript text omitted]